# Mitochondrial Genetic Disorders: Cell Signaling and Pharmacological Therapies

**DOI:** 10.3390/cells8040289

**Published:** 2019-03-28

**Authors:** Fatima Djouadi, Jean Bastin

**Affiliations:** Centre de Recherche des Cordeliers, INSERM U1138, Sorbonne Université, USPC, Université Paris Descartes, Université Paris Diderot, F-75006 Paris, France; jean.bastin@inserm.fr

**Keywords:** inborn mitochondrial disorders, pharmacological therapy, BZ, RSV, AMPK, NAD, PPAR, PGC-1alpha, SIRT1, ROS

## Abstract

Mitochondrial fatty acid oxidation (FAO) and respiratory chain (RC) defects form a large group of inherited monogenic disorders sharing many common clinical and pathophysiological features, including disruption of mitochondrial bioenergetics, but also, for example, oxidative stress and accumulation of noxious metabolites. Interestingly, several transcription factors or co-activators exert transcriptional control on both FAO and RC genes, and can be activated by small molecules, opening to possibly common therapeutic approaches for FAO and RC deficiencies. Here, we review recent data on the potential of various drugs or small molecules targeting pivotal metabolic regulators: peroxisome proliferator activated receptors (PPARs), sirtuin 1 (SIRT1), AMP-activated protein kinase (AMPK), and protein kinase A (PKA)) or interacting with reactive oxygen species (ROS) signaling, to alleviate or to correct inborn FAO or RC deficiencies in cellular or animal models. The possible molecular mechanisms involved, in particular the contribution of mitochondrial biogenesis, are discussed. Applications of these pharmacological approaches as a function of genotype/phenotype are also addressed, which clearly orient toward personalized therapy. Finally, we propose that beyond the identification of individual candidate drugs/molecules, future pharmacological approaches should consider their combination, which could produce additive or synergistic effects that may further enhance their therapeutic potential.

## 1. Mitochondrial Energy Metabolism Disorders 

The inborn defects of mitochondrial fatty acid β-oxidation (FAO) and respiratory chain (RC) rank among the most frequent genetic disorders of energy metabolism in human [1]. Although both groups of diseases are often presented and discussed separately by clinicians and researchers, they share many common features. Among them: (i) both groups of disorders are caused by the partial or total loss of function of a single protein or enzyme essential to the mitochondrial RC or to the FAO pathway; (ii) a large number of disease-causing genes have been identified, and each individual disorder is generally associated with a wide panel of gene mutations, with poorly understood genotype–phenotype correlations; (iii) the pathogenesis of these diseases involves not only bioenergetics defects due to disruption of mitochondrial structure and function, but also multiple other mechanisms—including ROS overproduction, oxidative stress, and accumulation of noxious intermediates; (iv) these disorders are clinically heterogeneous and can manifest in neonates, children, or adults with very diverse symptoms affecting one or several organs with high energy demand—including heart, skeletal muscle, liver, and brain; and (v) most of these disorders remain without treatment to date. Most importantly, and as we will see in this review, the FAO pathway and mitochondrial RC can be regulated, directly or indirectly, by the same transcription factors or co-activators, which leads to eventually propose the same therapeutic approach for both groups of disorders. Over the past few years, it has become clear that disruption of cellular bioenergetics contributes to the pathogenesis of many chronic or age-related common disorders, and this has strongly stimulated the research of drugs targeting mitochondrial energy production [2,3,4,5,6,7,8,9]. Data published on this topic shows that a number of candidate-based approaches presently under study for the treatment of common disorders have been hardly envisaged in the field of inborn mitochondrial disorders, and would definitely warrant further investigation. 

Given the large unmet medical demand in the field of mitochondrial genetic disorders, many new experimental therapies are presently developed, which cannot all be detailed in this review. Here, we chose to focus on pharmacological approaches based on small molecules targeting the mitochondrial bioenergetics. Emphasis was given to the proposed mechanisms of action and signaling pathways underlying the effects of these molecules, mostly based on patient cells and animal model studies. For the presentation of other strategies, including nutritional approaches, or for extensive data regarding clinical trials, the reader is referred to recently published excellent reviews and articles [10,11,12,13].

### 1.1. Fatty Acid Oxidation Disorders

Inborn mitochondrial FAO disorders represent a large group of rare diseases caused by mutations in nuclear genes encoding proteins or enzymes involved in import or catabolism of fatty acids in the mitochondria. The enzymology of mitochondrial FAO is well described and has been the topic of several reviews [14,15]. Very briefly, after being imported into the mitochondria, the long chain acyl coenzyme A (CoA)s enter the ß-oxidation pathway and are shortened by sequential removal of two carbons as acetyl CoA. Acetyl CoA is then fully oxidized to CO_2_ in the tricarboxylic acid cycle (TCA) cycle. Reduced nicotinamide adenine dinucleotide (NADH) and reduced flavin adenine dinucleotide (FADH_2_) produced from FAO are re-oxidized by respiratory chain at the level of complex I and electron transferring factor dehydrogenase (ETF-DH) respectively, culminating in the production of adenosine triphosphate (ATP) with high energy efficiency 

Clinical presentations vary from fatal multi-organ failure in newborns to adult-onset milder phenotypes [14,16,17,18]. Besides management of clinical manifestations, diagnosis of FAO disorders is also carried out as part of expanded newborn screening programs in several countries [14,17,19,20,21,22]. Screening is based on analysis of acylcarnitines in the newborn dried blood spots using tandem mass spectrometry, since the accumulation pattern of these metabolites provides a signature of the various FAO enzyme deficiencies. In this review, we will focus on some FAO defects (Figure 1) that are presented below.

Carnitine palmitoyl transferase 2 (CPT2) and very long chain acylCoA dehydrogenase (VLCAD) deficiencies are autosomal recessive long-chain FAO defects that are considered as relatively common. CPT2 forms with its counterpart, the CPT1, and the carnitine acyl-carnitine translocase (CACT), a shuttle allowing the entry of long chain acyl CoA in the mitochondria. The CPT2 is a monomeric enzyme associated with the inner mitochondrial membrane, encoded by a single gene located on chromosome 1p32 [23]. More than 60 CPT2 gene mutations have been described so far [24], associated with at least three distinct clinical presentations, which greatly differ in severity, age of onset, and tissue distribution. The most severe CPT2 presentation has been described only in very few instances in human fetuses, which exhibited brain and kidney dysgenesis, with neuronal migration defects [25]. This is in line with recent data suggesting that fatty acids might be an important oxidative fuel during embryonic development in neural cells [26,27]. Furthermore, some studies show that pharmacological inhibition or conditional deletion of the CPT system alter neural stem cell homeostasis in embryonic brain, suggesting that FAO might have an instructive role in neural cell differentiation [26,28,29]. CPT2 deficiency can also manifest in the neonate with severe intractable cardiac arrhythmia associated with hypoketotic hypoglycemia, potentially leading to Reye-like syndrome, with a high mortality. Finally, the most frequent phenotype of inborn CPT2 deficiency presents as an isolated skeletal muscle myopathy with onset varying from infancy to adulthood, characterized by myalgia, muscle weakness, and episodes of rhabdomyolysis triggered by exercise, and various other exacerbating stress conditions (fever, starvation). Each phenotype of the disease is associated with specific gene mutations and major differences exist between the mild and severe phenotypes with regards to the residual FAO capacities. Thus, tritiated long chain FAO rates measured in lymphoblasts or fibroblasts from severely affected patients are barely detectable, in contrast to those measured in cells from myopathic patients, which exhibit up to 40–50% of normal value. Residual CPT2 enzyme activity is also higher in the mild phenotype but, due to the difficulty to accurately measure very low levels of enzyme activity, overlaps can sometimes be found between the mild and the severe phenotype with regards to residual CPT2 activity [25]. CPT2 deficiency is one of the first inborn mitochondrial disease for which a relatively good correlation could be established between the severity of the clinical presentation and the amplitude of the metabolic defect, as reflected by patient cells studies. Importantly, this supports the notion that the CPT2 gene mutations associated with a mild disease phenotype do not fully disrupt the mitochondrial long-chain FAO capacities. 

VLCAD, 1 of the 11 mitochondrial acylCoA dehydrogenase isoforms, is a homodimeric enzyme associated with the inner mitochondrial membrane, encoded by a single gene on chromosome 17p13 [30]. The VLCAD catalyzes the first step of long-chain fatty acid β-oxidation, and in human, in contrast to rodent, this enzyme provides the bulk of long-chain acylCoA dehydrogenase activity in tissues with high FA utilization, with a negligible contribution of the LCAD isoform [31]. To date, a large number (>150) of disease-causing mutations, spread over the 20 exons of the VLCAD gene, have been identified, with at least three main phenotypes [14,32]. Briefly, the disease can manifest early after birth with severe life-threatening symptoms possibly associating cardiac arrhythmia, hepatic failure, and encephalopathy, with poor prognosis. An infantile less severe form also exists, mainly characterized by episodes of hepatic coma. Finally, as for the CPT2 deficiency, the commonest form of the disorder presents as an adolescent-onset myopathy with muscle weakness and myalgia, exposing the patients to a high risk of rhabdomyolysis following exercise or various other triggering factors, and possibly associated with progressive cardiomyopathy [18]. Diagnosis of VLCAD deficiency is also carried out as part of expanded newborn screening programs in several countries [20,21]. The newborns found positive by screening are often asymptomatic [33,34], but the long-term outcomes in these individuals cannot be predicted with certainty [35,36]. Accordingly, it is admitted that inborn VLCAD deficiency encompasses a continuum of severity ranging from a very low risk to develop symptoms up to life-threatening presentations. Genotype–phenotype correlations have been established only for the most severe forms frequently associated with "null" mutations (large deletions, frameshift, etc.) that fully disrupt gene expression [37]. However, the most common disease-causing DNA variations are missense mutations, with generally unpredictable effects on enzyme synthesis, stability, and catalytic activity, which can be found associated with all the disease phenotypes. In this disorder, some reports suggest that the levels of residual FAO flux measured in the patients’ cells might correlate with the clinical severity [32,38,39]. Of note, in a limited number of patients with the muscular form of CPT2 or VLCAD deficiency, the FAO defect is not expressed in the patient fibroblasts. 

Downstream the VLCAD, the mitochondrial trifunctional protein (MTP), embedded in the inner mitochondrial membrane, is structurally and functionally one of the most complex enzymes in the FAO pathway. Indeed, MTP is a hetero-octameric enzyme composed of four alpha and four beta subunits encoded by the *HADHA* and *HADHB* genes located on chromosome 2p23, which catalyzes the three final steps of long-chain fatty acid β-oxidation [40] (Figure 1). Inborn defects affecting the MTP can translate in isolated long-chain-3-hydroxyacyl-CoA dehydrogenase (LCHAD) deficiency (generally associated with *HADHA* gene mutations), or in decreased levels of the three MTP-borne enzymes activities (generally associated with *HADHB* gene mutations) [14,18,41]. Patients with MTP deficiency display a wide spectrum of clinical manifestations, from severe neonatal presentation with life-threatening cardiomyopathy, up to mild peripheral neuropathy with episodes of rhabdomyolysis [42,43]. The disease is particularly difficult to diagnose and characterize at the biochemical level, and, except in the case of the common G1528C transversion in the *HADHA* gene, the genotype–phenotype correlations are poorly defined.

### 1.2. Respiratory Chain Deficiencies

The inborn RC deficiencies form a large group of genetic disorders, caused by a biochemical dysfunction of one, or more, of the five hetero-multimeric RC complexes (CI to CV) (Figure 1) embedded in the inner mitochondrial membrane, which make up the oxidative phosphorylation (OXPHOS) machinery. These RC complexes comprise a total of 90 structural protein subunits, 13 of which encoded by the mitochondrial DNA (mtDNA), and associate several tens of ancillary proteins, encoded by the nuclear genome (nDNA), which are indispensable for their assembly, anchoring or regulation. Overall, more than 1000 nuclear-encoded proteins are required for the biosynthesis and functioning of healthy mitochondria. Pathogenic mutations in mtDNA can affect OXPHOS structural subunits or mitochondrial protein synthesis (tRNA, rRNA), while nuclear gene mutations can cause defects in OXPHOS function or assembly, but also in mtDNA expression and maintenance, or mitochondrial fusion or fission, which will severely affect OXPHOS organization and ATP production by mitochondria [44,45,46]. At the moment, more than 250 disease-causing nuclear genes have been listed, and their number is likely to augment with the rise of next generation sequencing/whole exome sequencing (NGS/WES). Mitochondrial diseases are the most common inborn metabolic disorders, affecting at least 1:5000 individuals, and are among the most common inherited neuromuscular disorders [1,45,46,47]. Clinically, they are very heterogeneous and can affect any organ or tissue, with almost any age of onset and course, varying from fatal neonatal to mild adult forms [45,47]. There is a general lack of genotype–phenotype correlations in most mitochondrial disorders [1]. CI and CIV deficiencies are the most frequent cause of inherited mitochondrial RC diseases.

Complex I (NADH:ubiquinone oxidoreductase) is the largest of all RC complexes, with an estimated molecular mass close to one megadalton. It comprises 46 subunits encoded by the nuclear (39 subunits) and mitochondrial (7 subunits) genomes, and catalyzes the transfer of 2 electrons from NADH to ubiquinone, coupled to the translocation of four protons from the mitochondrial matrix to the inter-membrane space. The human CI structure is only partially elucidated. It is organized in a catalytic arm that protrudes into the matrix and harbor the electron input and output (N and Q) modules, and a membrane arm for proton translocation (P module), embedded in the mitochondrial inner membrane [48]. The CI catalytic core consists of seven hydrophobic protein subunits (ND1 to ND6, ND4L) encoded by mtDNA, and of seven hydrophilic subunits consisting of two flavoproteins (NDUFV1 and NDUFV2) and five iron-sulfur protein subunits (NDUFS1, NDUFS2, NDUFS3, NDUFS7, and NDUFS8) encoded by nDNA. The biosynthesis and functioning of this complex requires assistance by at least fourteen ancillary/assembly factors [49,50,51]. Isolated CI deficiency ranks among the most common RC disorders, and comprises a number of clinically heterogeneous disorders of energy metabolism [1,49,51]. In some patients, biochemical characterization reveals different patterns of low molecular weight CI subcomplexes in cultured fibroblasts, together with a reduction in the levels of intact complex I, indicating that the mutations affect CI assembly and/or stability [52]. Thus, the biochemical diagnosis allows differentiating assembly/stability versus catalytic defects in CI deficient patients. Overall, highly variable biochemical and clinical features are observed among CI-deficient patients [50,53,54]. Of note, mild disease presentations, such as isolated myopathy with exercise intolerance, are clearly associated with significant residual levels of CI enzyme activity and assembled CI in the patient cells [55]. It was shown that fibroblasts from individuals with severely reduced CI amount and activity exhibited fragmented mitochondria, whereas normal mitochondrial morphology was found in moderately deficient patient cells [56]. At the molecular level, disease-causing mutations have been found in both nDNA and mtDNA [50]. A number of mutations have been identified in 19 of 39 structural subunits and in 10 of 14 assembly factors, with a majority of private non-recurrent mutations (reviewed in [1,49,51]). The disease is characterized by a broad phenotypic variability [53,54]. Mutations in mt-DNA encoded CI-subunits appear generally associated with mild phenotypes [55]. Nevertheless, the majority of patients carry mutations in nuclear-encoded CI subunits genes, possibly associated with severe neonatal presentations—including hypotonia, psychomotor retardation, lactic acidosis, and leukoencephalopathies—with poor prognosis. A high proportion of children ultimately develop Leigh syndrome and will die before the age of two [50,53].

The RC Complex IV or cytochrome c oxidase (COX) is a 200 kDa hetero-oligomer composed of 14 subunits. Three large hydrophobic subunits (COX1, COX2, and COX3 subunits), encoded by the mitochondrial genome, form the catalytic core, whereas the eleven others (COX 4, COX5a, COX5b, COX6a, COX6b, COX6c, COX7a, COX7b, COX7c, COX8, and NDUFA4), nuclear-encoded, are involved in the assembly and regulation of the enzyme [57,58]. The mammalian COX exists as monomers, dimers, and supercomplexes with RC complexes I and III [59]. COX holoenzyme activity requires several prosthetic groups including two hemes (a and a3), two copper centers (Cua and Cub), and zinc and magnesium ions [57]. As terminal component of the respiratory chain, COX catalyzes the transfer of electrons from cytochrome c to oxygen, coupled to the translocation of two protons from the matrix to the inter-membrane space. Assembly process and biogenesis of the enzyme is a complex multistep process requiring more than 30 nuclear-encoded accessory proteins that are not found in the final complex [57,60,61]. Assembly factors are required for stabilization and membrane insertion of nascent polypeptides, for synthesis of prosthetic groups, and for delivery of the metal cofactors. Among accessory proteins with clinical implications are COX10 and COX15, (heme biosynthesis), Surf1 (function still elusive), and LLPPRC, Sco1 and Sco2 (chaperones for copper delivery) [57,60,61]. As for CI, blue native gel electrophoresis allows measuring the steady-state levels of COX subunits and of sub-assembly intermediates. Assembly of COX has been characterized in numerous cell lines from patients with defects in assembly proteins. Biochemical studies in patient myoblasts or fibroblasts indicate a broad spectrum of changes in COX enzyme activity among various COX-deficient patients, with the most profound deficiency in patients with SURF1 mutations [59]. They also show that disruption of COX holoenzyme integrity triggers several mitochondrial dysfunctions, including decreased ATP content and membrane potential, and reduced cell growth with accelerated senescence [62]. COX deficiencies represent a heterogeneous group of disorders affecting predominantly tissues with high-energy demand, especially brain, muscle, and heart. Isolated COX deficiencies can arise from mutations in one of the three mt-DNA encoded subunits genes, or, more rarely, of mutations in one of the ten nuclear-encoded subunits genes, resulting in a variety of mild (myopathies with ragged-red fibers) or severe (encephalopathy, lactic acidosis, and stroke-like episodes) phenotypes [57,63]. However, the majority of affected patients harbor mutations in nuclear genes encoding COX assembly proteins, with variable associated phenotypes. Severe clinical symptoms often develop at birth or in the early childhood, such as encephalomyopathy, MELAS (mitochondrial myopathy, encephalopathy, lactic acidosis, and stroke-like episodes) syndrome, failure to thrive, and cardiomyopathy, possibly resulting in Leigh syndrome with necrotic bilateral brain lesions. Prognosis in children with COX deficiency is generally unfavorable, with very limited therapeutic options, and about half of them die in early childhood [59].

In recent years, the development of powerful technologies such as NGS/WES has undoubtedly revolutionized the diagnosis of inborn metabolism disorders. However, the management of affected patients is still challenging and too often limited to treatment of complications and supportive care, due to the lack of effective therapies for most of these disorders. Since the characterization of these devastating diseases, many compounds, vitamins, and drugs have been tested, often empirically, and with mitigated success, and there is to date no approved drugs for inborn RC-deficiencies. Besides the difficulty to set up clinical trials in this highly heterogeneous family of rare diseases, the question arises to know which therapeutic hypotheses and approaches should be tested at the pre-clinical step in order to increase the likelihood of success when moving to the clinical stage. As a matter of fact, high-throughput screening of compound libraries, or molecules’ discovery by serendipity, has not been up to the medical expectations to discover possible therapeutic compounds [64,65,66]. In this review, we have chosen to focus on the therapeutic hypotheses that are inspired from the basic knowledge on fine-tuning and molecular regulation of FAO and RC pathways, which underlies the concept of mitochondrial drugs and mitochondria-targeted therapy [2,8,67]. The interest in the quest of ‘mitochondrial drugs’ is not limited to inborn diseases and has considerably grown in recent years. Indeed, more and more studies lend support to the hypothesis that mitochondrial dysfunctions could contribute to the physiopathology of many common metabolic, cardiovascular, or neurodegenerative disorders, even though there is yet no drug licensed for the modulation, improvement, or correction of mitochondrial functions [2,3,4,5,6,7,8,9]. Accordingly, the data gathered from cell-based experiments in the context of inborn RC or FAO disorders might contribute to better evaluate the potential of drugs or natural compounds to cope for dysfunctions in basic mitochondrial energy production pathways, and to shed light on their mechanisms of action. 

## 2. Bezafibrate

Fibric acid derivatives, developed in the 80s, are widely prescribed hypolipidemic drugs [68]. Their mechanism of action was elucidated in the 90s, with the identification of peroxisome proliferator activated receptors (PPARs) as targets of these drugs. PPARs are ligand-activated transcription factors that belong to the nuclear receptor super-family [69]. Upon activation, PPARs bind as obligate heterodimers with the retinoid X receptor (RXR) to specific recognition sequences, called PPAR-response elements (PPRE), in the gene regulatory region of target genes, leading to cis-activation of gene transcription. The PPAR family includes three isoforms (α, β/δ, and γ) that exhibit different ligand specificities, tissue distributions, and physiological functions. PPARα is expressed at high levels in the liver, heart, and skeletal muscle, and plays a crucial role in the regulation of fatty acid catabolism by upregulating genes encoding enzymes in the β-oxidation pathway [70,71,72]. PPARα is activated by a variety of natural ligands including long-chain fatty acids, and by fibrates. PPARβ/δ is more widely expressed and has a 10- to 50-fold higher level than PPARα and γ in the skeletal muscle, in which it has been shown to play a key role in the regulation of energy metabolism [73,74,75]. Finally, PPARγ is highly enriched in adipose tissue and is also expressed in placenta and macrophages. PPARγ is activated by specific prostanoids and by thiazolidinediones, a novel class of antidiabetic drugs. In adipose tissue, it serves as an essential regulator for adipocyte differentiation and promotes lipid storage in mature adipocytes by increasing the expression of several key genes in this pathway [76,77]. 

### 2.1. Bezafibrate and Fatty Acid Oxidation Disorders

Positive effects of bezafibrate (BZ) in FAO-deficient patient cells were first established in the case of CPT2 deficiency, a disorder that typically illustrates the genetic and phenotypic complexity of FAO defects. The starting idea to test fibrates emerged from clinical and biochemical data on CPT2 deficiency and from fundamental knowledge on FAO regulation, available at the beginning of 2000. Thus, as mentioned above, it was found that the clinical severity of CPT2 defect was related to the severity of the metabolic block, as assessed by residual enzyme activity and FAO flux in patient fibroblasts. Second, basic molecular and in vivo data consistently suggested that many β-oxidation enzymes were transcriptionally regulated by PPARs [69] and fibrates were known to be agonists of these nuclear receptors. We hypothesized that fibrates could, via the PPAR nuclear receptors, stimulate CPT2 gene expression and residual enzyme activity, and could in return improve or correct the FAO rate in CPT2-deficient cells.

To address this question, fibroblasts from patients with mild or severe CPT2-deficiency (Table 1) were treated in parallel with each of the four different molecules of the fibrate group: BZ, fenofibrate, ciprofibrate, and gemfibrozil, commonly used for the treatment of hyperlipidemia. Typically, none of the fibroblasts derived from severely affected patients responded to fibrates. Furthermore, BZ was the only fibrate capable to upregulate FAO in fibroblasts from mildly affected patients, via an activation of *CPT2* gene expression [78], later confirmed by other authors [79]. Subsequent studies in myoblasts from patients with the muscular form of the disease confirmed that exposure to BZ restored palmitate oxidation to similar values as those measured in control myoblasts [80]. In parallel, exposure of myoblasts to high-affinity PPAR ligands revealed that correction of CPT2 deficiency was only achieved after treatment with a PPAR delta agonist, while a PPAR alpha agonist had no effect. Thus, CPT2 gene expression is under specific control of PPAR delta isoform in human muscle cells, and this explained why, among the four tested fibrates, BZ was the only efficient one. Indeed, BZ is known to bind the alpha and delta PPAR isoforms, whereas the other fibrates are strict PPAR alpha agonists [81,82,83]. Interestingly, CPT2 activity was not restored to normal values after treatment by BZ, however the drug-induced increases in residual enzyme activity appeared sufficient to allow correction of FAO flux [78]. This is consistent with the fact that carrier individuals harboring one mutated allele exhibit normal FAO rates despite reduced (50% or less) CPT2 enzyme activity. BZ treatment was also shown to lower the accumulation of toxic C16-acylcarnitine in patient fibroblasts and myoblasts [80,84,85]. Altogether, these data provided ex-vivo evidence for correction of mild inborn CPT2 deficiency using drugs targeting the PPAR signaling pathway. 

In an extension of these studies, the effects of BZ were investigated in fibroblasts from patients with the muscular form of VLCAD deficiency. Initially, BZ was shown to upregulate VLCAD mRNA, protein level, and enzyme activity, and could correct palmitate oxidation values in fibroblasts from three patients with the myopathic form of the disease, whereas no effects were observed in fibroblasts from severely affected neonates [86]. To better characterize the potential of BZ, a comparative study was then performed in 36 fibroblasts cells lines with distinct genotypes. About one-third of these cell lines displayed barely detectable palmitate oxidation rates and faint VLCAD western-blot signals and enzyme activities, unchanged after exposure to BZ. These fibroblasts all originated from patients with severe phenotypes. The 21 other cell lines, which exhibited variable deficiencies in the absence of treatment, responded to BZ by highly significant increases in FAO, often resulting in the restoration of normal values. Altogether, the pattern of responses could be explained by variable increases in residual enzyme activity in response to BZ, which, in turn, could be ascribed to variable consequences of mutations on enzyme stability/activity [39,87]. Partial correction of palmitate oxidation rates, as seen in some VLCAD-deficient fibroblasts, likely reflects the persistence of a functional FAO deficiency despite treatment by BZ, which could be confirmed by acylcarnitines analysis [39]. Additional studies of BZ in VLCAD-deficiency were performed by other authors, based on the analysis of acylcarnitine profiles in fibroblasts from patients representative of the various phenotypes [84,88]. The results indicated potentially beneficial effects of BZ, reflected by decreases in long-chain acylcarnitines with concomitant increases in acetyl-carnitine, a marker of β-oxidation flux. In some cases, however, it appeared difficult to distinguish the responses from mild versus severe patients’ fibroblasts. This might be due to intrinsic variability of acylcarnitine determination compared to the palmitate oxidation test. Indeed, newborn screening of VLCAD deficiency, which relies on the detection of specific acylcarnitines, suffers a high rate of false positive values, and recent studies suggest that FAO flux assays provide a better index to stratify risk in presumably positive newborns [38]. Altogether, VLCAD gene is transcriptionally upregulated in human fibroblasts exposed to BZ, and this can form the basis to improve or correct FAO deficiency in cells harboring various inborn VLCAD-deficiencies. 

Subsequently, we studied the effects of BZ in twenty-six fibroblasts cell lines from diagnosed MTP-deficient patients harboring *HADHA* or *HADHB* gene mutations, and in control fibroblasts. In this panel, a majority of patient fibroblasts appeared profoundly FAO-deficient in the tritiated palmitate assay, and were little or not improved after treatment by BZ [89]. Five cell lines reached FAO levels comprised between 60 and 80% of normal values, and only one cell line was restored to control, after exposure to BZ. In patients with *HADHA* gene mutations, western blot analysis revealed decreased levels of both *HADHA* and *HADHB* gene products, suggesting that the biosynthesis and assembly of the multimeric MTP enzyme was negatively impacted by the presence of aminoacid substitutions on one only of its subunits. FAO inductions were associated with modest but significant increases in residual LCHAD and LCKAT enzyme activities in response to BZ. Altogether, the effect of BZ greatly varied from one individual to another one, likely due to variable effects of the mutations on the assembly and activity of this FAO enzyme. Gene expression and protein levels of HADHA and HADHB, together with the LCHAD and LCKAT enzyme activities, were found inducible by BZ in control fibroblasts, and most likely contributed to the upregulation of FAO capacities in treated versus untreated control fibroblasts. Altogether, the aforementioned studies suggest that at least one of the enzymes involved in the mitochondrial import of long-chain fatty acids (CPT2), and the 4 enzymatic steps involved in their β-oxidation (VLCAD and MTP), are upregulated in human control fibroblasts, in response to BZ. 

The molecular mechanisms that might account for the effect of BZ on FAO pathway are depicted Figure 2. Thus, BZ can bind directly PPARα and/or PPARδ, which after activation and heterodimerization with RXR directly bind PPAR response-elements present in almost all FAO genes. Interestingly, this transcriptional activation of FAO genes is likely ‘boosted’ by an additional mechanism involving PGC-1α. PGC-1α (PPARγ co-activator 1α) is a transcriptional co-activator that does not bind directly DNA, but interacts with and co-activates several important transcription factors involved in energy metabolism such as PPARs, estrogen-related receptors (ERR), the nuclear respiratory factors 1 and 2 (NRF1/2), or TFAM [90,91]. PGC-1α is also well known to play a central role in the coordinate regulation of nuclear and mitochondrial genes, essential for mitochondrial biogenesis and function. Remarkably, a PPAR response element (PPRE) [92,93] and an ERR binding site [94] have been described on the promoter region of PGC-1α gene, which form the basis of an auto-regulatory feed-forward loop, allowing to amplify its own transcription. Therefore, after activation of its gene expression, PGC-1α can co-activate PPAR, as well as ERRs. The ERRs are a group of orphan receptors, which control hundreds of genes involved in mitochondrial function including a subgroup of FAO genes that overlaps with PPAR targets [77,95,96].

### 2.2. Bezafibrate and Respiratory Chain Disorders

The question arose whether this new pharmacological strategy could be extended to RC disorders since, like in FAO disorders, residual enzyme levels of deficient RC complex can be detected in some cases of RC defects. The ‘rationale’, however, was not obvious because, no PPREs had ever been found in the promoter regions of OXPHOS genes, although data obtained in rodents clearly suggested that the PPAR signaling pathway could regulate some RC genes [103,104]. This led to surmise an indirect mechanism likely involving major molecular regulators of mitochondrial RC, such as NRF, Tfam and PGC-1α, as detailed below.

Initial experiments established that exposure of control human fibroblasts to BZ induced upregulation of several nuclear genes encoding subunits or ancillary proteins of RC CI, CIII, or CIV. In parallel, the corresponding RC complexes enzyme activities were significantly augmented as well as the rates of cellular pyruvate- and succinate-driven respiration in polarographic assays, in response to BZ [105]. Thus, BZ had a strong stimulatory effect on the activity of RC complexes and on cellular oxygen consumption rates in control fibroblasts. When added to the culture medium of RC-deficient fibroblasts, BZ was found to potentially exert similar effects, inducing in particular, significant increases in the protein levels and enzyme activities of deficient RC complex I, III, or IV in 9 out of 14 cell lines tested (Table 2), and the normalization of cellular oxygen consumption rates in some treated cells. The effects of BZ on CI and CIV activity could be mimicked by treatment of control and patient cells with the PPAR delta agonist GW0742, while the PPAR alpha agonist had no effect. Consistent with the presumed mode of action of BZ, positive effects of treatment were only achieved in cell lines that exhibited some level of residual RC function, i.e., when the disease causing mutations did not lead to highly unstable or absent mutant protein [105]. Further studies from other groups confirmed that BZ could upregulate the activities of RC complex I, III, and IV in human skin fibroblasts, and displayed similar effects in a variety of cell lines including HeLa, HEK293, bone marrow-derived cells, and human brain astrocytoma cells [106,107,108]. In a compared study of patients’ fibroblasts harboring mutations in *Sco2* or *Surf1 gene*, it was found that treatment with BZ could correct COX activity in the Sco2-deficient cell line, which exhibited partial COX deficiency, but not in the Surf1 mutant, which was almost devoid of residual enzyme activity [107]. Subsequent studies in five cybrid cells harboring different mitochondrial tRNA mutations, including those responsible for the MELAS or the MERFF syndrome, showed that treatment with BZ upregulated the affected tRNAs and the mitochondrial protein synthesis, increased overall OXPHOS enzyme activity, and improved aerobic ATP production [109]. The effects of BZ in RC-deficient patient cells (100µM, 72h) were also investigated at first in six fibroblasts cell lines harboring mutations in different nuclear-encoded Complex I subunits or assembly factors [110], and then in four cell lines from patients with multiple RC deficiencies harboring mutations in different nuclear encoded components of the mitochondrial translation machinery [111]. These authors concluded on possibly favorable, cell-line specific, effects of BZ on growth rate, ROS production, ATP content, and mitochondrial membrane potential. However, no conclusion could be drawn concerning a possible correction of RC-defects by BZ since residual enzyme activity and mutant protein levels were not investigated, nor were cellular oxygen consumption rates.

From a mechanistic point of view, we showed that in controls fibroblasts, BZ induced the expression of several genes encoding subunits or assembly proteins of RC complexes. We next found that PGC-1α NRF1/2, and Tfam mRNAs were also increased by BZ treatment [105]. These results led us to propose the following signaling cascade (Figure 2) to account for the effects of BZ on mitochondrial RC. Thus, after activation by BZ, PPAR transcriptionally upregulates the levels of PGC-1α, which, in turn, acts as co-activator of PPAR and ERR nuclear receptors forming the feed-forward mechanism, already described above for FAO, that greatly amplifies the biological activity of PGC-1α. It has also been reported that the NRF2 gene expression was directly regulated by ERRs [95,96]. ERR and NRFs transcription factors could, thereby, trigger a coordinate transcriptional upregulation of numerous RC nuclear genes and of Tfam [128], which controls the transcription and replication of mitochondrial genome. Importantly, since ERRs are involved in the regulation of almost every aspect of mitochondrial functions and mitochondrial biogenesis [96], a possibly massive recruitment of ERRs associated with increased levels of its co-activator PGC-1α could, in the end, considerably scale up the mitochondrial biogenesis program. This proposed regulatory cascade could explain in what manner the PGC-1 co-activators are central to the stimulation of mitochondrial biogenesis, which relies on a coordinate regulation of nuclear and mitochondrial genomes.

The potential of BZ has also been evaluated in vivo in several mouse models of COX deficiency: i.e., a *Surf1* constitutive knockout (KO) mouse [129], a muscle-specific *Cox15* KO mouse [129], the “Deletor” mouse overexpressing a mutant replicative helicase Twinkle [130], and the Mutator mouse expressing a mutant mtDNA polymerase γ [131]. In these experiments, mice received BZ added to standard diet (0.5%) for one to eight months. Briefly, *Surf1*^−/−^ mice exhibited no apparent phenotype, with in particular normal motor performance, but showed a mildly decreased COX enzyme activity in the muscle. This is in contrast with humans in which, as aforementioned, *Surf1* mutations lead to severe COX deficiency associated with progressive encephalopathy in infancy. COX activity or RC-related genes were not modified by BZ treatment, nor were mtDNA content, citrate synthase or activities of other RC complexes [129]. By contrast, the muscle-specific *Cox15*^−/−^ mice displayed both physical and biochemical hallmarks of a COX-deficient mitochondrial myopathy, with, for instance, a severe COX deficiency in muscle, in parallel with proliferation of abnormal mitochondria, which translated in reduced motor performance in treadmill test. In this muscle-specific *Cox15*^−/−^ model, BZ worsened the mitochondrial myopathy [129]. The Deletor mouse represents a good model for human late-onset mitochondrial myopathy, since the overexpression of mutant Twinkle leads to accumulation of multiple mtDNA deletions and progressive apparition of COX-negative fibers. BZ administration to Deletor mice delayed the progression of the late-onset adult-type myopathy with a diminished amount of COX-negative fibers and a decreased mtDNA load. BZ did not change COX enzyme activity and did not induce mitochondrial biogenesis [130]. Although not exactly a model of mitochondrial disorders, the mutator mouse exhibited increased mtDNA mutations associated with marked mitochondrial dysfunction, and developed many features of premature aging. Treatment with BZ induced FAO but not mitochondrial content or function in skeletal muscle, whilst improving some premature aging-like phenotypes [131]. Overall, these studies indicated little changes of respiratory chain complexes and of gene expression, and of mitochondrial content/function in the skeletal muscle of BZ-treated mice, while the expression some FAO-related genes was increased. However, some treated animals developed severe hepatomegaly with major lipid utilization alterations, reflecting rodent-specific toxic effects of high doses of fibrate [132]. These unwanted side effects prevented any definite conclusions on possible effects of BZ on RC deficiency in these models. Of note, in another study in mTOR knockout mice presenting with progressive myopathy, it was shown that chronic treatment by BZ partially restored the activity of COX and of succinate dehydrogenase in mTOR-deficient muscle, and upregulated gene expression of PGC-1α and of several mitochondrial genes including CPT1, MCAD, and LCAD, as well as COX and citrate synthase [133]. BZ was also tested, but at much lower doses, in Taffazzin knockdown (*Taz*) mice [134], a model of human Barth syndrome, in which major alterations of mitochondrial oxidative metabolism were found [135]. The *Tafazzin* gene encodes a mitochondrial transacylase involved in the synthesis of cardiolipin, a unique mitochondrial phospholipid crucial for the integration of electron transport chain supercomplexes in the inner mitochondrial membrane, and for their interactions with FAO enzymes. *Taz* mice were shown to exhibit progressive dilated cardiomyopathy with left ventricular dysfunction [135], and reduced exercise performance [134]. The knockdown of tafazzin induced inactivation and destabilization of RC supercomplexes (SC) and decreased gene expression of mitochondrial FAO enzymes and RC complexes in cardiac muscle [134]. More recently, Jang et al. reported a more detailed characterization of heart dysfunction in *Taz* mice. In particular, the authors showed that in addition to the loss of SC integrity leading to decreased CI, CII and CIII activities, *Taz* heart showed high basal but low Ca^++^ -induced mitochondrial swelling and mitoROS, associated with high levels of cyclophilin D, a key regulator of mitochondrial permeability transition pore opening [136]. After four months on a pelleted rodent chow containing 0.05% BZ, *Taz* mice exhibited improved systolic function, but intriguingly did not ameliorate their exercise capacity. Importantly, RNA-seq analysis revealed drug-induced upregulation of several genes related to mitochondrial energy metabolism (FAO, RC, TCA) in the cardiac muscle [134]. 

BZ is a widely used hypolipidemic drug with more than 30 years of therapeutic experience and a good safety profile. We thus performed in 2006, an open-label clinical trial of BZ (3 × 200 mg BZ/day) in six symptomatic CPT2-deficient patients over 6 months [97], followed by a period of 36 months of treatment continuation [137]. Briefly, the FAO [97] and oxygen consumption rates [132] were found increased in isolated muscle mitochondria of all the patients after six months of treatment. The protein levels of CPT2 [137] as well as NDUFV1, COX2, and COX4 [132] were enhanced by treatment with BZ. Studies in patient myoblasts also showed that the FAO capacities were markedly enhanced in response to BZ in the six genotypes. Clinically, all the patients reported improvements in daily physical activity, and muscular pain, as well as in quality of life. These studies were followed by two others clinical trials that reported either no positive, or variable effects of BZ on patients’ conditions. Orngreen et al. reported no improvement in FAO and heart rate during cycle test, and unchanged Borg scores and acylcarnitines in five CPT2 and five VLCAD patients (no genotypes reported) receiving BZ [138]. This appears in contrast with our results, however, the patients enrolled in this trial presented a very mild phenotype compared to ours, since, for instance, no muscle symptoms or rhabdomyolysis episodes were reported during the six-month trial period. Finally, the most recent trial was performed in two CPT2 and six VLCAD patients receiving BZ at various dosages during six months [139]. In this trial, no data were provided regarding the in vivo or ex vivo FAO capacities in the panel of patients, before or after treatment by BZ. The effects of BZ on rhabdomyolysis and muscular pain were generally variable, but of note all patients reported an improved quality of life. In the three trials, increased carnitine plasma levels were found and no adverse effects were reported. Overall, we still think that BZ alone or in combination (see Section 4, and Conclusions), could be helpful to improve the condition of some patients, although the reasons of such variability among the various clinical trials are not totally clear. Based on this, some issues for reflection can be suggested, which may be useful to improve future clinical trials on CPT2 or VLCAD disorders. At first, screening ex vivo response to drugs in the patients’ fibroblasts should be envisaged as a pre-requisite in the process of inclusion. Inclusion of patients with very mild symptomatology should be avoided. Physical activity on day-to-day basis could nowadays be monitored using high technology accelerometers with computerized data analysis. Daily muscular pain and other patient-reported outcomes, monitored by e-diary, could also represent more relevant endpoints. Finally, the classical biomarkers such as acylcarnitines or CPK plasma levels appear too variable to be considered as good endpoints, and more elaborate markers based on metabolomics should be developed.

## 3. Resveratrol

The history of resveratrol (RSV) is very interesting, indeed the plants in which this polyphenol is naturally present, have been known for more than 1500 years in traditional Asian medicine books, to be “good for muscle, bone, and longevity”. RSV (3,5,4’-trihydroxystilbene) was first chemically isolated from the roots of white Hellebore (*Veratrum grandiflorum O. Loes*) in 1940 in Japan [140], and thereafter in 1963 from the roots of *Polygonum cuspidatum* [141], a plant described as a prescription for "inflammation, carcinogenesis, cardiovascular diseases and longevity”. Since then RSV is endowed with the same benefits than the medicinal herbs from which it was isolated. Thus, this phytoalexin synthesized by plants in responses to various stresses is surmised to have anti-oxidant, anti-inflammatory, and anti-proliferative properties, with possibly favorable effects in cancers, metabolic, cardiovascular and neurodegenerative diseases. In human food, the presence of RSV in red wine is often mentioned to explain the ‘French Paradox’, an expression coined to describe the observation that the French population seems to exhibit a relatively low risk of cardiovascular disease despite consumption of many foods rich in saturated fat. This is generally attributed to the possible protective effects of regular red wine consumption. RSV is presently tested in many common disorders in a large number of clinical trials (ClinicalTrials.gov). It was further demonstrated by two different studies published in the field of ‘diabetes and obesity’ [142,143], that dietary supplementation of RSV to high-fat fed mice protected against obesity and improved running endurance. In these mice studies, part of the beneficial effects of RSV was attributed to increases in skeletal muscle mitochondrial density, oxygen consumption and oxidative-type fibers, induced by RSV [144]. This prompted us to probe for possible positive effects of RSV in mitochondrial diseases, and in particular in mitochondrial myopathies.

### 3.1. Resveratrol and Fatty Acid Oxidation Disorders

Dose–response and kinetic experiments in control and deficient fibroblasts showed that RSV induced an increase of FAO flux, already significant at 10–20 µM after 24 h of treatment, and maximal with 75 µM RSV for 48 h. We then studied nineteen fibroblasts originating from severely or mildly affected CPT2 or VLCAD-deficient patients (Table 1). Fibroblasts treated, with 75 µM RSV for 48 h exhibited markedly enhanced FAO rates in cell lines with mild FAO deficiency, i.e., the myopathic form of CPT2 or VLCAD deficiency, leading to restore normal flux, while no effect was found in fibroblasts from patients with the severe form of the deficiencies [100]. Interestingly, we found that the magnitude of FAO increases triggered by this natural compound (RSV) were as large as those induced by BZ. Likewise, we showed that RSV induced the relief of the metabolic bottleneck through significant increases in the amount of mutant CPT2 or VLCAD protein. We then further evaluated the effects of other stilbenes (cis-RSV and piceid) naturally found associated with RSV (trans-RSV) in food, and of the major human RSV metabolites (RSV-3-O-glucuronide, RSV-4-O-glucuronide, RSV-3-O-sulfate and di-hydro-RSV) [99]. Cis-RSV and piceid as well as the RSV metabolite produced by intestinal microbiota, i.e., di-hydro-RSV were the most efficient to stimulate FAO flux (between +30% to +130%). Altogether, these data suggest that trans-RSV, some of its metabolites, and other stilbenes might act in concert to stimulate FAO in human, and, remarkably, could improve mild CPT2 or VLCAD deficiency in patient fibroblasts.

### 3.2. Resveratrol and Respiratory Chain Disorders

These encouraging results led us to extend our studies to RC disorders, specifically in CI- or CIV-deficient fibroblasts harboring mutations in nuclear-encoded subunits or assembly factors (Table 2). As in the case of FAO disorders, we took care to include fibroblasts from patients with severe or mild CI or CIV deficiencies, to further evaluate the importance of the defect severity in the response to RSV. These experiments showed that in moderate CI- or CIV-deficient fibroblasts, treatment with RSV was able to increase the mutant protein levels and residual enzyme activities. In some cell lines, these increases, though relatively modest, were sufficient to correct the RC defect, as evidence by the O_2_ consumption values. These favorable effects of RSV were also confirmed by the improvement of lactate/pyruvate ratio values in the treated cells [116]. Interestingly, we observed in control fibroblasts that RSV coordinately induced many proteins (structural subunits or assembly factors) of the five RC complexes and upregulated the O2 consumption rates. Like BZ, and as could be anticipated, RSV did not increase residual enzyme activity, hence did not improve RC deficiency, in fibroblasts exhibiting severe RC-deficiencies. These data are in line with observations made by other authors, in which the lack of beneficial effects of RSV in patient fibroblasts harboring different RC defects could clearly be related to the severity of the disease presentation [110,111,122,124]. 

While many authors reported that RSV could stimulate mitochondrial biogenesis in animal models [142,143,145,146,147,148], the data obtained from human cells were, in comparison, scarce. In this regard, we found that the indexes of mitochondrial population, assessed by Mitotracker green staining, citrate synthase activity or Tfam protein levels, were all increased in response to RSV in control fibroblasts, and in the mildly deficient fibroblasts in which a correction of RC defect was observed. This might emphasize the importance of mitochondrial biogenesis in the correction of RC deficiency, however other observations clearly bring to qualify this conclusion. Indeed, evidence for a stimulation of mitochondrial biogenesis was also obtained from fibroblasts with severe RC deficiency treated by RSV, in which a profound enzyme deficiency still persisted, and no improvement of O_2_ consumption were observed [116]. This is in agreement with the results of De Paepe et al., showing that no correction of CII or CIV deficiency occurred after RSV treatment of severe RC-deficient fibroblast, even though an increase in citrate synthase enzyme activity was observed [122]. Activation of mitochondrial biogenesis without correction of RC deficiency is also supported by several studies, which consistently reported increases in mitotracker green staining intensity when switching cultured patient fibroblasts from a standard to a glucose-free medium supplemented with galactose [110,111,126]. This medium change is known to dictate an increased dependence over mitochondrial oxidative metabolism to produce ATP, and in line with this, appears to provoke an adaptive increase in mitochondrial biogenesis. Despite this, the passage of patient fibroblasts in glucose-free medium does not lead to improve or correct the RC-deficiency. 

Altogether, these data support the notion that stimulation of mitochondrial biogenesis is, by itself, not sufficient to alleviate inborn mitochondrial deficiencies, and this underlines an important point of discussion concerning the effects of RSV, or of other molecules, like AICA riboside (see Section 5.3) in RC- or FAO-deficient cells. Thus, possibly beneficial effects appear to be primarily dependent on whether the functional bottleneck in the RC or FAO pathway can be alleviated, or not. Indeed, and as illustrated Figure 3, correction will not be reached if the mutations result in undetectable enzyme activity, often associated with the most severe clinical presentations, even though a mitochondrial biogenesis might occur in response to the pharmacological treatment.

To our knowledge only two reports on the effects of RSV on RC defects due to mtDNA mutations are available. The first one tested an unusual protocol of RSV treatment by different concentrations applied daily for four days to fibroblasts carrying the following mutations m.3243A>G in MT-TL1 gene, m.8344A>G in MT-TK gene or m.8993T>G in MT-ATP6 gene [149]. The authors found that treatment with 0.01µM RSV for 24 h increased oxygen consumption (OCR) and ATP production. However, although most increases were found significant, the values of OCR and ATP after treatment of deficient fibroblasts did not reach control levels. In the second study, the authors reported NMR-based metabolomics analysis of fibroblasts from CI-deficient LHON patients (m.11778G>A) after treatment with 50 µM of RSV for 24 h. They found that LHON fibroblasts exhibited specific metabolic alterations, in particular, increased intracellular lipid levels and decreased amino acids levels that were partly reversed by RSV treatment. However, possible improvements in mitochondrial bioenergetics were not documented [150].

Since therapy of mitochondrial disorders should benefit of new knowledge on the regulation of mitochondrial energy metabolism, we sought to investigate the molecular mechanism(s) involved in the observed effects of RSV. Over the last few years, there has been a consensus about the main participants of RSV signaling cascade in various cells and animal models, with SIRT1, and/or AMPK and/or PGC-1α being generally considered as chief actors. An oversimplification of this cascade is shown Figure 4. Briefly, many authors reported that RSV activated AMPK [142,151,152] and SIRT1 [143,148,153], although different molecular mechanisms underlying these activations were proposed [148,152]. AMPK for AMP-activated protein kinase (see Section 5) is widely recognized as a cellular sensor of metabolic stress and energy deprivation, due to its ability to phosphorylate key enzymes, transcription factors and co-activators, including PGC-1α. SIRT1 is one of the member of the sirtuins family of NAD^+^-dependent protein deacetylases comprising seven members, which regulate several aspects of cellular and mitochondrial physiology. The sirtuins play key roles in modulating the acetylation status of mitochondrial proteins, transcription factors and regulatory signaling proteins, as well as histones [154,155]. SIRT1 has attracted considerable attention as a potential pivotal regulator of mitochondrial biogenesis, due to its capacity to sense changes in NAD^+^ levels (see Section 7), and to deacetylate/activate PGC-1α in return [156,157]. Concerning the action of RSV, some authors proposed that AMPK activation increased the NAD^+^ level, which would lead to SIRT1 activation. Accordingly, PGC-1α could ultimately be activated both by phosphorylation via AMPK, and by deacetylation via SIRT1. However, there are still uncertainties on these issues, and in recent years, several authors reevaluated various aspects of RSV signaling in animals and cell models and questioned in particular the role of SIRT1 [158,159,160,161,162].

In our first study of the effects of RSV (75 µM) on FAO defects, the use of siRNA against PGC-1α demonstrated its involvement in the RSV signaling cascade in human fibroblasts [100]. Next, in order to evaluate the implication of SIRT1, we first used 10 mM nicotinamide (NAM) to inhibit SIRT1 in the cultured fibroblasts and, surprisingly, found that NAM did not prevent the RSV-induced increase in FAO rates (unpublished data). This lack of effect could eventually be interpreted as a species difference since this dose had been successfully used by several authors to inhibit SIRT1, but in rodent cells exclusively. We then turned to another SIRT1 inhibitor, namely sirtinol (40 µM), considered at that time as a specific SIRT1 inhibitor but more recently identified as a pan-sirtuin inhibitor. Accordingly, experiments performed in the presence of sirtinol led us to erroneously conclude that SIRT1 was involved in mediating the effects of RSV on FAO-deficient fibroblasts. Later experiments using the same RSV concentration (75 µM) in RC-deficient fibroblast demonstrated that specific knockdown of SIRT1 with siRNA only marginally decreased the pharmacological response, thereby demonstrating that the effects of RSV were not mediated by SIRT1 [116]. Altogether, this led us to revise our initial conclusion on the role of SIRT1 in mediating the effects of RSV in FAO-deficient fibroblasts. Another argument supporting the lack of involvement of SIRT1 came from experiments using the specific SIRT1 activator SRT1720, which never increased the cellular FAO capacities when tested at 50, 100, 250, and 500 nM for 48 h in control and in FAO-deficient fibroblasts (FD and JB unpublished data). Overall, one can conclude that in human fibroblasts, treatment by high concentrations of RSV i.e., 75 µM, stimulated mitochondrial energy metabolism mainly via a SIRT1-independent signaling pathway. Finally, it has been suggested that the involvement of SIRT1 could be dose-dependent. Indeed, in the rodent C2C12 cell line, a moderate dose of 25 µM RSV was found to stimulate mitochondrial metabolism in a SIRT1-dependent manner, while the effects observed at 50 µM RSV appeared SIRT1-independent [148]. 

A possible role of AMPK in mediating RSV effects was also investigated in RC-deficient fibroblasts by use of siRNA, and this demonstrated that like SIRT1, AMPK did not participate to the molecular signaling in human fibroblasts treated with 75 µM RSV. Therefore, the signaling pathways involving SIRT1 and AMPK such as proposed in Figure 4, could not account for the positive effects of RSV that were observed in the patient fibroblasts. These unexpected results led us to search for other targets of RSV, and we turned to estrogen receptors (ERs), because RSV is known to be a phytoestrogen capable to bind ER [163]. Of note, the estrogen receptors could coordinately regulate mitochondrial and nuclear RC gene transcription, either directly or indirectly via relevant downstream targets such as ERRα, PGC-1α, and Tfam [164]. The use of ER (ICI182780) and ERR (XCT790) specific inhibitors demonstrated that both receptors played a major role in the signaling cascade mediating the RSV effects in human fibroblasts. Finally, our results are in agreement with the assumption that RSV has more than one direct target in the cells [158,165], and the following scheme to explain the effects of high concentrations of RSV on FAO and RC in human fibroblasts can be proposed (Figure 5).

In addition to RSV, we tested two other phytochemicals, either derived from traditional herbal medicines—i.e., berberine [166]—or present in food and beverages—such as quercetin. However, neither berberine nor quercetin elicited any significant effect on FAO flux in human control or FAO-deficient fibroblasts (FD and JB personal data).

## 4. Effects of BZ + RSV Combinations

The marked inductions of residual FAO capacities in response to BZ or to RSV prompted us to address the effects of combined treatments associating both compounds. In a first set of experiments, CPT2- and VLCAD-deficient fibroblasts (myopathic form) were treated in parallel for 48 h with the doses of each compound used in previous experiments, i.e., 75 µM RSV, or 400 µM BZ, or with the combination of those (75 µM RSV + 400 µM BZ), and the palmitate oxidation rates were then measured (Figure 6). 

As expected, the FAO deficiencies were fully corrected in the CPT2-deficient cells after exposure either to 75 µM RSV or to 400 µM BZ, however, even higher levels were obtained in cells treated with RSV + BZ. In the VLCAD-deficient panel of cells, the potential of BZ or of RSV alone to upregulate FAO was also confirmed, with amplitudes of FAO inductions in response to each molecule often larger than those observed in CPT2-deficient fibroblasts. Exposure to RSV or to BZ alone did not fully correct the FAO deficiency in four out of seven VLCAD-deficient patient cell lines. In contrast, treatment with RSV + BZ allowed to restore normal FAO rates in all but one of the VLCAD-deficient fibroblasts, and induced spectacular increases in some cell lines, for instance in the VLCAD deficient patient (p.Lys382Gln/p.Glu130fsX216), in which FAO capacity rose from 0.8 nmol FA/h/mg protein in vehicle-treated up to 5.19 nmol FA/h/mg protein (x6.5) in doubly-treated cells. As in the case of CPT2-deficiency, the effects of both compounds together were always larger than those of each individual compound. As depicted in Figure 6, these results emphasize the ability of endogenous cellular signaling pathways to trigger upregulation of FAO in response to combined pharmacological treatments, which trigger remarkable FAO increases. 

In subsequent experiments, we further analyzed the patient cell responses to dual treatments, but with special focus on the effects of low concentrations of RSV and BZ [167]. Briefly, initial experiments were performed in p.Val283Ala/p.Val283Ala VLCAD-deficient patient fibroblasts treated in parallel for 48 h with 16 different combinations of RSV (10, 20, 37.5, or 75 µM) plus BZ (25, 50, 100, or 400 µM) in the culture medium, prior to FAO determination (FD and JB unpublished data). FAO correction was actually reached with eight different RSV + BZ combinations that associated submaximal concentrations of RSV (<75 µM) and BZ (<400 µM). This could be explained not only by additive, but also by synergistic effects of RSV+BZ, which were apparent in the lower range of concentrations tested [167]. An illustration of this low-dose synergy in CPT2- and in VLCAD-deficient fibroblasts is presented in Figure 7. 

In the four cell lines, a 48 h treatment by a combination of 30 µM RSV plus 35 µM BZ induced dramatic increases (from +93% in p.Val283Ala/p.Val283Ala to +252% in p.Ala304Thr/p.Gly439Asp) in β-oxidation flux, and resulted in the restoration of FAO capacities in the control range. In contrast, as can be seen, the effects of treatment with 35 µM BZ only, were quite modest (from +6% to +21%), and were never found significant, in any of the four cell lines. Treatment by 30 µM RSV alone increased FAO (from +43% in p.Val283Ala/p.Val283Ala to +106% in p.Ala304Thr/p.Gly439Asp), but to a much lower extent compared to the dual treatment. Accordingly, the amplitude of FAO stimulation induced by exposure to 30 µM RSV + 35µM BZ significantly exceeds the value that could be expected from fibroblasts treated with 30 µM RSV only, or with 35 µM BZ only. At the end, the FAO values after treatment with 30 µM RSV plus 35 µM BZ were equivalent to those reached after treatment with a single molecule at much higher concentrations, i.e., with 400 µM BZ or with 75 µM RSV (not shown). Altogether, these results suggest that low concentrations of RSV and BZ could activate distinct signaling mechanism in a feed forward-loop resulting in potentiation of their effects on FAO in the treated fibroblasts.

One hundred and fifty-three clinical trials are presently testing RSV in healthy people or in various conditions such as type 2 diabetes, obesity, heart failure, Alzheimer or cancer. Of note, one trial (ClinicalTrials.gov Identifier: NCT03728777) is recruiting to test RSV in mitochondrial myopathies and FAO defects. Given the results presented above, the combination RSV + BZ could be worth testing. 

## 5. AMPK Activators

The rationale of testing drugs or compounds known to target key regulatory factors of mitochondrial energy metabolism, naturally led to focus on the AMP-activated protein kinase (AMPK), considered to be the fuel gauge of the cells.

### 5.1. AMP-Activated Protein Kinase (AMPK)

Maintaining sufficient levels of ATP is crucial to sustain proper biological functions of all living cells. As a consequence, the eukaryotic cells have developed mechanisms to match energy demand with energy supply. The AMPK is well known to play a very important role in this homeostasis. Indeed, AMPK is considered to be a cellular energy sensor capable to adapt catabolic versus anabolic processes, and to adjust the energy status of the cell to its environment [168,169]. AMPK is a heterotrimeric Ser/Thr kinase composed of three non-identical protein subunits. The α subunits (α1 or α2) are the catalytic subunits and contain the Thr-172 residue, whose phosphorylation is necessary for a fully active enzyme, as explained below. The β (β1 or β2) and the γ (γ1, γ2, γ3) subunits are the regulatory subunits [168].

AMPK activation generally occurs in situations in which ATP synthesis is compromised (starvation, hypoxia, ischemia) or when ATP consumption is accelerated (muscle contraction, exercise). In these situations, AMPK is activated by the fall in ATP occurring concomitantly with a rise in AMP and ADP. Indeed, in cells with normal glucose and oxygen supplies, a high ATP/AMP ratio is maintained by adequate catabolism, and the reaction catalyzed by the adenylate kinase keeps the AMP concentration very low. However, any metabolic stress (in particular glucose or oxygen deprivation) that leads to decrease the ATP supply will displace the equilibrium towards AMP production. Therefore a rise in AMP level instructs the cell that the energy balance is compromised, which activates AMPK [168,169]. Indeed, the binding of AMP stimulates AMPK activity by two mechanisms (Figure 8): (1) AMP causes a quick activation of AMPK (up to 10-fold) through an allosteric mechanism (2) more importantly, AMP binding promotes the phosphorylation of the subunit at Thr-172 by upstream kinases (LKB1 or CaMKK) and reduces the rate of dephosphorylation of AMPK by protein phosphatases, resulting in a >100-fold activation. Altogether, the combination of allosteric and phosphorylation effects triggered by AMP leads to a >1000-fold activation of AMPK. In the current model, LKB1 is constitutively active and continuously phosphorylates AMPK at Thr-172, which, in the absence of AMP, is immediately dephosphorylated. Activation of CaMKKs by Ca^2+^ calmodulin provides an alternate pathway to activate AMPK in response to increased intracellular Ca^2+^ levels, independently of changes in AMP [168,169].

As already mentioned, once activated in response to energy shortage, AMPK switches on catabolic pathways (to produce ATP) while simultaneously switching off anabolic pathways (ATP consumers). To do so, AMPK at first mediates short-term effects, through direct phosphorylation of metabolic enzymes, as well as long-term effects, through control of transcription to adapt gene expression to energy demand. Thus, acute activation of AMPK increases glucose uptake by skeletal muscle through the translocation of GLUT 4 to the membrane, stimulates glycolysis, and decreases glycogen synthesis through direct phosphorylation of glycogen synthase and 6-phosphofructo-2-kinase, respectively. Altogether, activation of AMPK therefore rapidly mobilizes glucose in ATP-generating pathways. In parallel, AMPK acutely induces FAO, by means of the phosphorylation and inactivation of acetyl-CoA-carboxylase-2 that produces malonyl-CoA, an allosteric inhibitor of CPT1, leading to promote the entry of long chain fatty acids into the mitochondria for their β-oxidation [170]. These acute effects of AMPK do not appear relevant in the context of therapy, since short-term phosphorylation events could not conceivably cope for genetic loss of function of enzymes in the FAO pathways.

More worthwhile for this review, it has also been shown that AMPK was able to regulate transcriptional programs through phosphorylation events [171], and in particular could mediate adaptive long-term increases in mitochondrial energy production. For example, and as reviewed in [171], chronic treatment of mice with AICA riboside, a classical AMPK activator, increases gene expression of mitochondrial genes and induces a mitochondrial biogenesis in skeletal muscle. All these effects are abolished in animal models with impaired AMPK activity. In addition, studies using genetically modified animals, whether AMPK KO mice or mice overexpressing active or inactive forms of AMPK, clearly confirmed that AMPK acts as a master transcriptional regulator of genes involved in mitochondrial energy metabolism [171]. 

The effects of AMPK activation on mitochondrial genes are very likely mediated through the activation of several transcriptional activators and co-activators. AMPK is known to directly phosphorylate/activate target nuclear transcriptional regulators such as Forkhead Box O (FOXO) [172] and cAMP response element-binding proteins (CREBs) [173] (Figure 9). Of note and as already mentioned, AMPK is a key regulator of the co-activator PGC-1α [171]. Accordingly, a strong link between AMPK and PGC-1α is suggested by studies based on pharmacological or transgenic/KO strategies [174]. This has been confirmed by the observation that AICA riboside was unable to stimulate mitochondrial energy metabolism in muscle, in the absence of PGC-1α [175]. The same study demonstrated that AMPK could directly interact with and phosphorylate PGC-1α in vitro, at Thr^177^ and Ser^538^, leading to PGC-1α activation. However, it was also suggested that this PGC-1α phosphorylation does not directly increase the intrinsic co-activator activity but rather primes it for its subsequent deacetylation and activation by SIRT1 [156]. This precise point is still debated in the literature but does not question the fact that AMPK seems to be a potent regulator of PGC-1α activity [174,176]. Furthermore, it was shown that AMPK activation increases PGC-1α expression [175], likely through PGC-1α feed-forward regulatory loop [93,177]. Of note, in addition to ERR and PPAR, several binding sites for transcription factors such as CREB and myocyte enhancer factor 2 (MEF2), are also present in the PGC-1α promoter. Figure 9 summarizes the effects of AMPK on some transcription factors and on PGC-1α at transcriptional and post-translational levels.

These molecular mechanisms likely account for the long-term effects of AMPK activators on mitochondrial energy metabolism and on mitochondrial biogenesis. Altogether, these data argue for a pivotal role of AMPK in the fine regulation of mitochondrial oxidative metabolism. Recently and most interestingly, the role of AMPK has been extended to other aspects of mitochondrial biology.

### 5.2. AMPK and Mitochondrial Homeostasis

In the cells, mitochondria form a dynamic network, which undergoes permanent fusion and fission. These processes are now thought to be essential to adapt to various situations—such as stress or high-energy demand—and for the degradation of damaged mitochondria. Recent studies uncovered an important role of AMPK in the regulation of mitochondrial homeostasis though its action on key proteins involved in mitochondrial fission and mitophagy [178]. First, it was shown that in human cells, inhibition of RC with rotenone or antimycin led to AMPK phosphorylation and to the fragmentation of the mitochondrial network. These authors also showed that AMPK was required to mediate mitochondrial fission [179]. Furthermore, the findings that direct activation of AMPK with small molecules was sufficient to induce mitochondrial fragmentation clearly posit the AMPK as a direct and major regulator of mitochondrial dynamics. The mitochondrial fission factor (MFF) was then identified as a novel substrate of AMPK [179]. MFF, located in the mitochondrial outer membrane, is the primary receptor for dynamin-related protein 1 (DRP1), which catalyzes mitochondrial fission. Finally, AMPK also directly phosphorylates ULK1 (Unc-51 autophagy activated kinase 1), a kinase involved in autophagy and mitophagy [178], two highly dynamic processes responsible for removal of dysfunctional cellular components. Thus, AMPK is not only the fuel gauge of the cells but seems to be endowed with remarkable capacities to regulate the mitochondrial homeostasis. 

For these reasons, the AMPK signaling pathway could be a highly relevant system in the search of new therapeutic targets for the correction of inborn mitochondrial disorders. Before this, AMPK has attracted much attention, over decades, in the therapy of common diseases [180]. Thus, major efforts have been made by academic groups or pharmaceutical companies to search and develop AMPK activators, with different mechanisms of action [181]. Briefly, AMPK can be activated: i) indirectly, by compounds inhibiting ATP synthesis, such as metformin; ii) by compounds converted to AMP analogs, such as AICA riboside; iii) by allosteric activators that directly bind AMPK at sites different from the AMP binding sites (see A769662 or GSK773); and iv) by oxidative stress.

### 5.3. AMPK Activators and Fatty Acid Oxidation Disorders

AICA riboside: Much of our knowledge about the cellular functions of AMPK and its target genes comes from the use of the chemical ‘tool’ AICA riboside (5-aminoimidazole-4-carboxamide-1-β-ribofuranoside) a cell-permeable nucleoside, erroneously abbreviated as AICAR in a large number of published studies [182]. AICAR is in fact the established acronym for AICA riboside also called ZMP, an intermediate of the de novo purine nucleotide biosynthesis. After entering the cells, AICA riboside is converted by phosphorylation into ZMP, a mimetic of 5’-AMP that activates AMPK. Thus, ZMP binds to AMPK at the same sites as AMP. Interestingly, AICA riboside (also called Acadesine) is a drug tested in chronic lymphocytic leukemia [183] and in myocardial ischemic injury [184]. 

Five CPT2-deficient fibroblasts and 5 VLCAD-deficient fibroblasts were treated with 500 µM of AICA-riboside for 48 h (FD and JB unpublished data). Surprisingly, the treatment did not lead to increase the FAO flux but rather tended to significantly decrease palmitate oxidation in two out of five CPT2-deficient cell lines and in all the VLCAD-deficient fibroblasts tested. These results therefore indicated that long-term treatment with high concentrations of AICA riboside might lead to inhibition of FAO in human fibroblasts. Although unexpected, these results are in agreement with the study of Guigas et al., which clearly showed that AICA-riboside also inhibits cellular mitochondrial respiration in isolated rat hepatocytes by an AMPK-independent mechanism that likely results from the combined intracellular inorganic phosphate depletion and ZMP accumulation [185]. Taken together, these data support the warnings of many researchers concerning the role of AMPK in mediating the AICA riboside effects, i.e., that some cellular effects of this compound could be AMPK-independent. Indeed, it is now well acknowledged that AICA riboside is not a specific AMPK activator and, in line with this, interpretation of results obtained with this compound in studies in which no attempts were made to silence AMPK in order to prove its direct involvement, should be taken with caution [182].

A769662: The first direct AMPK activator described was A769662 from Abbott laboratories. As mentioned above, this compound is particularly interesting since it binds directly and specifically a unique site in the β1 subunit, which differs from the AMP binding site. A-769662 is thus an allosteric activator that does not work as an AMP mimetic, capable to activate AMPK independently of the energy status of the cells. A-769662 also protects Thr172 from dephosphorylation by protein phosphatase [186]. In a study published in 2011, we treated six different CPT2- or VLCAD-deficient fibroblasts with three concentrations of A769662 (30, 60, and 100 µM). No significant changes in palmitate oxidation were measured up to 100 µM [100]. Subsequently, we increased the concentrations tested in the dose–response and 300µM was chosen as the most effective dose, without cell toxicity. Using these conditions, we then assessed the effects of A769662 for 48h on FAO flux in controls and in a panel of patients’ fibroblasts with the mild form of CPT2 or VLCAD-deficiency (Figure 10). 

Treatment with A-769662 significantly stimulated (x1.2 to x1.7) FAO utilization in controls and in five cell lines, but resulted in the correction of FAO rates in only one cell line with mild CPT2 deficiency. The results were quite different with the nine VLCAD-deficient fibroblasts. The increases of palmitate oxidation in treated fibroblasts varied from 1.2- to 2-fold compared to vehicle-treated cells. None of these inductions resulted in the restoration of normal palmitate oxidation. Taken together, these results indicate that A769662 had variable moderate effects on FAO in the patients’ fibroblasts and was clearly less potent than other molecules like BZ or RSV. Furthermore, we think that this modest response to A769662 could be fibroblast-specific, since better results were obtained in deficient patient myotubes (FD and JB personal data). 

GSK773: Finally, the most recent data from our group explore the therapeutic potential of GSK773, another direct and specific activator of AMPK, which, like A769662, binds the β1 subunit [98]. The experiments were performed in myotubes of patients with the adult form of CPT2 deficiency, which is particularly interesting since this disorder presents as a metabolic myopathy. Of note, recent studies have suggested a pivotal role of AMPK in skeletal muscle plasticity [187,188,189,190]. AMPK was found constitutively activated in the patients’ myotubes, which displayed both a reduced FAO and, unexpectedly, an impaired differentiation process. Indeed, the Myosin Heavy Chain (MHC)-I, isoform expressed during differentiation in oxidative slow-twitch fibers and the MHC-IIA expressed in fast-twitch glycolytic/oxidative fibers were both found significantly downregulated in the CPT2-deficient myotubes, compared to controls. In addition, the fusion index, a well-admitted parameter to assess muscle differentiation process, was also decreased in the patients’ myotubes. In agreement with the mitochondrial dysfunction, and consistent with the basal phosphorylation of AMPK, the mitochondrial network was found fragmented. 

Treatment with GSK773 stimulated residual FAO capacities in a dose- and time-dependent manner and improved or corrected several hallmarks of FAO deficiency in the patients’ myotubes. Correction of CPT2 defect was achieved after treatment with 30 µM GSK773 for 48h, and this clearly involved upregulation of mutant of CPT2 protein levels. Analysis of acylcarnitines’ species in the culture media revealed, as expected, an accumulation of C16-acylcarnitines, which was significantly decreased after GSK773 treatment. Importantly, GSK773 also fully corrected the differentiation process and induced in particular a shift in the MHC isoforms toward the slow oxidative type in the CPT2-deficient myotubes. Upregulation of mitochondrial biogenesis most likely played an important role in the beneficial effects of GSK773, as indicated by studies of complementary indexes and by the improvement of the mitochondrial network quality. Indeed, mitotracker immunofluorescence studies revealed that exposure to GSK773 induced an elongation of the network, with increased mitofusin 2 (MFN2) levels, a marker of mitochondrial fusion, while the levels of DRP1, marker of mitochondrial fission, were decreased. We then sought to probe for the molecular mechanisms underlying the effects of GSK773 using siRNA knockdowns and pharmacological approaches. These studies revealed the implication of AMPK, PGC-1α, and p38 MAPK, as well as the central role of the ROS signaling pathway, all key players of skeletal muscle plasticity [187,188,189,190]. The proposed signaling pathways involved in the effects of GSK773 are summarized in Figure 11. It appears that the treatment with GSK773 recapitulates several important features of skeletal muscle adaptation to exercise, suggesting that this compound might represent an “exercise mimetic” [191]. Altogether, these results obviously argue for AMPK as a highly relevant therapeutic target for pharmacological correction of the muscular disorders associated with CPT2 deficiency.

### 5.4. AMPK Activators and Respiratory Chain Disorders

AICA riboside: In 2011, Golubitzky et al. reported the effects of seven different compounds including AICA riboside (500 µM, 72 h) on several parameters including cellular growth, ROS and ATP in six CI-deficient fibroblasts harboring mutations in nuclear genes encoding CI subunits [110]. In this study, controls and CI-deficient fibroblasts were cultured in glucose-free medium supplemented with galactose. As could be expected, NDUFS2-deficient cells exhibited decreased levels of ATP and lower growth rates compared to controls. However, surprisingly, immunofluorescence experiments revealed decreased phospho-AMPK steady-state levels in the patient fibroblasts, despite the proven energy deficiency induced by glucose deprivation. Treatment with AICA riboside was shown to significantly increase cellular growth and ATP levels. However, no increase in oxygen consumption was found that could account for this increase in ATP. The authors also concluded that AICA riboside increased mitochondrial biogenesis, but the experiments performed with mitotracker green or mitotracker red gave discrepant results, with either an increase or a decrease in the fluorescent staining, respectively. Finally, the beneficial effects of AICA riboside were not proven to be AMPK-dependent since only the expected phosphorylation of AMPK by AICA riboside was shown, without quantification of the total AMPK, and since no siRNA against AMPK were used. In another study, four others cells lines with mutations in nuclear genes encoding mitochondrial translational machinery were studied for their response to various pharmacological compounds [111]. Treatment with AICA riboside was found to exhibit variable effects and no clear conclusions could be drawn, since for instance it increased (i) COX/SDH ratio in GFM1 cells; (ii) ATP content in MRPS22 cells; and (iii) mitochondrial content in TRMU, by unknown mechanisms.

AICA riboside was also tested in vivo in three different mouse models of COX deficiency, in which BZ had also been tested, namely *Surf1*^−/−^, a constitutive *Sco2* knockout/knockin mouse (*Sco2^KO/KI^), ACTA-Cox15*^−/−^ [129]. The authors found that AICA riboside did not change mtDNA content and citrate synthase activity in the skeletal muscle. However, Tfam and COX subunits encoding genes were increased in all the treated mouse models, leading to parallel increases of COX subunit proteins, and ultimately resulting in enhanced COX enzyme activity. Finally, in two COX-deficient mouse models, these AICA riboside-induced changes translated in marked improvement of motor performance. More recently, the group of CT Moraes confirmed the beneficial effects of AICA riboside treatment in a mouse model of mild mitochondrial myopathy, in which a muscle-specific ablation of *COX10* was generated [192]. Interestingly, in this study, the treatment was given for three months in either pre-symptomatic or post-symptomatic animals. In these protocols, long-term AICA riboside treatment maintained or restored the running endurance of the animals, which could be attributed to improvements of CIV enzyme activity, without changes in the activities of other OXPHOS complexes. Of note, the lack of significant increases in citrate synthase, mtDNA and in various OXPHOS proteins led to conclude that the beneficial effects of AICA riboside were not primarily due to a global increase in mitochondrial biogenesis. This is in line with data showing that AICA riboside could have AMPK-independent effects, as aforementioned [182]. The authors next sought to determine if changes of muscle fibers types might account for the symptoms’ alleviation in the treated animals. However, it was found that AICA riboside increased the mRNA of MHCIIa only in wild-type animals. Transcriptome analysis led to conclude that AICA riboside induced muscle fibers regeneration, which seemed to be, in this model, the main mechanism responsible for the recovery of the myopathy. Finally, autophagy and unfolded protein response might also have contributed to the improvement of mitochondrial dysfunctions. Altogether, the experiments performed in mouse models of COX deficiency showed that treatment with AICA riboside improved several characteristic features of this mitochondrial defect, although the mechanisms underlying this beneficial effect seem to be quite different in the different models, and may not necessarily involve AMPK.

### 5.5. Others Compounds Tested 

Metformin: This biguanide known to be an indirect activator of AMPK is one of the front-line treatments of type 2 diabetes, already prescribed to more than 100 million patients worldwide [193]. Metformin has been proposed to activate AMPK by way of a mild and transient inhibition of RC CI, leading to slow down the ATP synthesis. Despite being introduced in the 50s, the exact mechanism of action of metformin is, however, still debated. In addition, some of its effects have been shown to be AMPK independent [193]. Nevertheless, the good safety record of this compound and its well-admitted effects as AMPK activator, prompted us to test its capacity to stimulate FAO. In our hands, and although metformin was found to phosphorylate AMPK in fibroblasts, treatment with 25 µM of Metformin for 48h did not modify the FAO capacities in CPT2- or VLCAD-deficient fibroblasts (JB and FD personal data).

Thiazolidinediones (TZD): Thiazolidinediones are a class of anti-diabetic drug, comprising troglitazone, pioglitazone, and rosiglitazone, which were primarily identified as high affinity ligands for PPARγ. In recent years, it was shown that TZD could also rapidly activate AMPK [194,195,196], and this incited us to test the three TZDs for their possible beneficial effects on FAO disorders. First, dose–response experiments (5, 15, 30 µM) were performed in control and FAO-deficient fibroblasts. A significant increase of FAO was detected at 5µM, reaching its maximal at 30 µM for all the TZDs. We then focused our experiments on the most effective one, namely pioglitazone (Figure 12).

Exposure to 30 µM pioglitazone for 48 h resulted in marked increases in FAO in control fibroblasts, and led to the correction of FAO in the four CPT2-deficient fibroblasts. By comparison, in VLCAD-deficient fibroblasts, treatment with pioglitazone exerted more variable effects, ranging from no change to a 1.6- or 1.9-fold increase in FAO flux. Correction of FAO was reached in two out of four VLCAD-deficient cell lines. Pioglitazone upregulated the level of VLCAD protein in control and in VLCAD-deficient fibroblasts (Figure 12C). In line with these results, Cha et al. showed that the depressed rate of palmitate oxidation measured in human myotubes of type 2 diabetes subjects, compared to controls, was significantly improved after chronic treatment (four days) with 10 µM pioglitazone [197]. Two possible mechanisms might account for the positive effect of pioglitazone on FAO. First, we showed by western blot that AMPK was phosphorylated by pioglitazone (30 µM, 48 h), in human fibroblasts (JB and FD personal data), in agreement with studies performed in muscle biopsies of type 2 diabetes patients showing that AMPK is also phosphorylated, after six months of treatment with pioglitazone [194]. Interestingly, these authors subsequently showed that pioglitazone upregulated the expression of several genes involved in FAO (*CPT1B, ACADM, HADH*) and in OXPHOS (*COX6C, NDUFA5*), as well as PGC-1α expression, in patients’ muscle. Nevertheless, the causal relationship between the AMPK activation and the FAO increase, or the stimulation of mitochondrial transcription, remains to be proven. The second mechanism by which pioglitazone might increase FAO is by cross-reacting with, and activating, PPARα. Indeed, the binding and transactivation assays of the three human PPAR (hPPAR) isoforms by pioglitazone showed that while pioglitazone is a selective hPPARγ agonist with an EC50 of 0.49 µM, it can also activate hPPARα at concentrations of 10µM and above [198]. Accordingly, one cannot rule out that the effects of pioglitazone observed in control or FAO-deficient cells were mediated through PPARα activation, and the exact mechanism of action could only be established by specific silencing of each PPAR isoform. 

Of note, it was reported that 12 weeks of treatment with pioglitazone induced an increase in mitochondrial respiration, and in the mitochondrial CII and CIII protein levels in skeletal muscle of type 2 diabetes patients, while rosiglitazone inhibited the mitochondrial respiration, indicating possibly important differences between the various TZD with regards to their effects on mitochondrial oxidative metabolism [199]. Pioglitazone (ACTOS, Takeda Pharmaceuticals) was approved by the FDA and by the European Medicines Agency (EMA) in 2000. In 2012, and although EMA had issued warnings about increased risks of bladder cancer in patients with type 2 diabetes receiving pioglitazone, this drug was not withdrawn from the market, and it is still used with some restrictions [200]. Altogether, this first set of data suggests that pioglitazone could represent another good candidate drug for the treatment of mitochondrial energy metabolism, although this requires carefully taking into account the risks and benefits of its use.

## 6. Antioxidants

One of the common concepts that has emerged concerning the physiopathology of mitochondrial disorders is the existence of cellular oxidative stress originating either from FAO or RC blockade, explaining the enthusiasm of many researchers to test the possible beneficial effects of antioxidants in various diseases’ models and in patients as well.

### 6.1. Antioxidants and Fatty Acid Oxidation Disorders

It is now admitted that the inborn FAO disorders not only affect the use of fatty acids as energy substrates but induce a global mitochondrial dysfunction due to multiple factors including alterations in CoA and carnitine metabolism, and accumulation of incompletely oxidized fatty acids and acylcarnitines [201,202,203]. Several studies performed in different rat tissues established that buildup of long- or medium-chain fatty acids derivatives was associated with oxidative stress induction and to a progressive depletion of cellular glutathione [204,205]. In line with this, studies in cultured fibroblasts from patients with SCAD, CPT2, LCHAD, or MCAD deficiency demonstrated a decreased cell survival, compared to controls, when challenged with the antioxidant menadione, and this could, at least partly, be alleviated by concomitant treatment with BZ, N-acetyl cysteine or vitamin C + vitamin E treatment [102]. Other mechanisms might contribute to cellular redox stress in cells with inborn FAO defects. It has been shown in particular that overexpressing the disease-associated SCAD mutant enzyme in astrocytes induced an increased ROS production [206], suggesting that in FAO defects, as in some neurodegenerative disorders, accumulation of misfolded protein might contribute to increase ROS production and oxidative damages [202]. Considering the large number of disease-associated genes responsible for FAO disorders, and the diversity of gene mutations encountered in the patient population, it is presently difficult to globally evaluate the importance of ROS-related mechanisms in the cellular damages and dysfunctions associated with inborn FAO deficiencies. Studies in patient fibroblasts appear particularly interesting in this respect since these cell lines reflect the genetic heterogeneity of FAO disorders, however these studies provided, so far, relatively variable results. In a recently published study, eight LCHAD- and one VLCAD-deficient fibroblasts cell lines were found to exhibit elevated ROS levels, measured by image analysis after staining with the CM-H2XROS fluorescent probe, which were corrected by treatment with a NADPH oxidase 2 (NOX) specific inhibitor, pointing to this enzyme as a major ROS production site in all the patient fibroblasts [207]. Interestingly, the mutations in mitochondrial trifunctional protein were also associated with changes in the organization of mitochondrial network, characterized by an increased fragmentation reverted after treatment with the NOX inhibitor. The FAO capacities were, however, not analyzed in this study, and it remains unknown whether manipulations that alleviate or correct ROS production could have any effects on FAO capacities. In a very recent study, Seminotti et al. examined in parallel the superoxide and the ROS status in three VLCAD-deficient fibroblasts cell lines, using the MitoSOX red and the CM-H2DCFDA probes, respectively [101]. Under standard culture conditions, the fluorescent intensities were only slightly superior to those obtained from control fibroblasts. However, after passage in glucose-free medium, the superoxide and ROS signals were amplified, and the differences between patient and control fibroblasts were found more significant. Intriguingly, treatment of VLCAD-deficient fibroblasts in glucose-free media with known antioxidant compounds (1mM N-Acetyl Cystein, 75 µM RSV, 200 nanoM MitoQ, 1mM Trolox) did not mitigate superoxide production, whereas treatments with the experimental JP4-039 or XJB-5-131 ROS scavengers were effective [101]. The fact that RSV did not decrease the ROS levels is in contrast with data obtained in our group (FD and JB unpublished data). Indeed, in four different experiments, measurements of total cellular ROS using the CM-H2DCFDA probe were performed in six VLCAD-deficient fibroblasts cell lines from patients with the myopathic disease phenotype. Of note, the fibroblasts were cultured in HamF10 medium containing glucose. Four out of six fibroblasts exhibited significant ROS overproduction. After fibroblasts’ treatment with 75 µM RSV for 48 h, the ROS levels were strongly diminished and returned to the levels found in control fibroblasts in most cell lines. Interestingly, since we previously showed that exposure to RSV also induced correction of FAO deficiency in these cell lines, as recalled in Section 3.1 [100], this natural molecule associates effective antioxidant as well as FAO upregulation properties.

### 6.2. Antioxidants and Respiratory Chain Disorders

The mitochondrial respiratory chain is one of the main cellular sites of superoxide production since, under normal conditions, a small proportion of electrons transferred along the respiratory chain complexes are diverted to react with oxygen and form superoxide radicals. This electron leakage can occur at the level of RC complex I and complex III, and of ETF-dehydrogenase [208]. It is now proven that ROS production at physiological levels plays important roles in cell signaling, whereas excessive ROS production could have complex deleterious consequences [209,210], by inducing accelerated degradation of mitochondrial proteins, oxidative modifications of cellular proteins, and membrane lipids peroxidation. Based on this, the hypothesis that some RC disorders might lead to ROS overproduction and cellular oxidative damages and that treatment by antioxidant might represent a valuable therapy has long been popular in the field of RC disorders. Furthermore, a number of natural or synthetic compounds with presumed antioxidant properties are available as food supplements, and generally well tolerated, and are therefore relatively easy to prescribe in human, which has been an additional incentive to directly assess their potential in RC-deficient patients. Accordingly, there is a significant amount of published data reporting case-to-case studies or clinical trials aimed at evaluating antioxidant compounds in patients with various RC-deficiencies, in contrast to the situation encountered in the case of many other candidate drugs, for which in vivo clinical are scarce or completely lacking. This has been the topic of many excellent reviews [10,11,12,13] and will not be presented here. It should be pointed out, however, that these in vivo trials provided generally variable results and were often inconclusive. Accordingly, the question to know whether antioxidants might be potentially useful in the repertoire of drugs to combat the manifestations of RC deficiencies is still debated. In view of this, different experimental models, and in particular patient fibroblasts, offer valuable tools to address the basic underlying questions on this topic. Indeed, fibroblasts allow evaluating the levels of ROS production, the activity of antioxidant defense system, and the extent of cellular oxidative damage, in relation to various forms of RC deficiency, and provide a relatively simple model to evaluate the potency of antioxidants, in parallel.

In RC-deficient patient fibroblasts, increased ROS production is well documented in the case of CI deficiency [112,113,114,117,119,126,211,212,213], and has also been reported in CIV [123] and CV deficiency [214]. Overall, abnormal ROS levels were found associated with a variety of mutations of nuclear or mitochondrial genes encoding RC complex subunits or assembly factors, as well as other mitochondrial proteins [112,215], and in CoQ10-deficient fibroblasts [216]. In these studies, the amplitude of ROS overproduction in RC-deficient fibroblasts, compared to fibroblasts from healthy individuals, was quite variable. Additionally, ROS production in patient fibroblasts did not appear to correlate either with the nature of the disease, or with the extent of RC deficiency, or with the severity of the phenotype. Importantly, RC deficiencies with normal ROS levels in patient fibroblasts are not infrequent [115,119,125,212,215,217]. At the cellular level, it is tempting to speculate that ROS-related oxidative stress could play a major role in triggering the mitochondrial dysfunctions and alterations of mitochondrial structure described in RC-deficient fibroblasts [112]. Nevertheless, the consequences of increased ROS production in terms of cellular oxidative stress are not clearly understood. Indeed, some studies suggest the existence of disrupted antioxidant protection and free radical damage to cellular lipids and mitochondrial proteins [119,213,214,215]. On the other hand, other data indicate no changes in cytoplasmic and mitochondrial thiol redox status and lipid peroxidation, and minor or absent oxidative stress in RC-deficient patient fibroblasts [123,125,211,215,218]. Overall, it is admitted that, in some patient cells, increased cellular ROS is a key feature of RC deficiency, and might therefore represent a target for treatment strategies, but a number of questions still remain. In particular, it is unclear to what extent alterations in ROS production might have an instructive role to promote cellular adaptations to RC deficiency, or, alternatively, should be considered as deleterious. Another question is to know whether these alterations in ROS production are upstream or downstream of structural and functional damages observed in mitochondria of RC-deficient cells. As mentioned above, several antioxidant compounds (lipoic acid, EPI-743, KH176, etc.) are presently under study in RC-deficient patients [11]. However, for some of them, this was performed in the absence of preliminary studies in patient cells. Hence, the mechanisms of action of antioxidants compounds are still poorly understood at the cellular level. The data available in-patient fibroblasts vary mainly by the nature of the antioxidant(s) tested, and by the cellular outcome(s) chosen to evaluate its potency. 

Quinone analogues: Coenzyme Q10 (CoQ10) is the most well-known quinone analogue since this natural lipid-soluble compound freely circulates within the inner mitochondrial membrane where, in its reduced form, it carries electrons between different complexes. This explains why CoQ10 has been tested in broad range of OXPHOS diseases, with mitigated success, except for patients with primary CoQ10 deficiency [219]. CoQ10 was tested in parallel with three other quinone analogues i.e., coenzyme Q1, decylubiquinone, and idebenone in four fibroblasts from LHON (Leber hereditary optic neuropathy) patients harboring the m.11778A>G mutation at homoplasmic levels. Cell growth, ROS production, Δψ, ATP content, ATP production and oxygen consumption rate were measured, and idebenone was reported to increase ATP production whilst reducing ROS levels in some cell lines [220]. Interestingly, in vivo idebenone is rapidly converted into its metabolites [221] and it was recently shown that one of them namely QS10 was able to partially restore mitochondrial respiration in CI-deficient fibroblasts with mutations in mt-ND1 gene [222].

N-Acetyl Cysteine (NAC): NAC is considered as a classical antioxidant, potentially increasing the activity of glutathione metabolism enzymes while directly interacting with radical species. In two studies, NAC supplementation (1-4 mM, 72 h) was shown to normalize ROS levels in three out of four cell lines harboring different mutations of nuclear genes encoding mitochondrial translation-related proteins [111], and in three fibroblasts cell lines with RC complex I or IV defects [126]. Improvements in ROS went together with slight positive effects of NAC treatment on cell growth and ATP content in glucose-free medium, but possible beneficial effects of NAC on some critical parameters, such as oxygen consumption, or RC residual enzyme activity were not investigated. In a recent study of three CI deficient patient cell lines, increased ROS levels were found associated with ACAD9- but not to ND6- or NDUFV1- deficiency, and treatment of ACAD-9 fibroblasts by 1 mM NAC for 24 h did not improve superoxide production [115]. In this study, by contrast, a new mitochondria-targeted antioxidant (JP4-039) was reported to decrease superoxide level and to increase basal and maximal mitochondrial respiration. Finally, in a study of two CI-deficient fibroblasts cell lines harboring mutations in the nuclear NDUFS8 or in the mitochondrial ND5 subunit genes, minor differences were found in the mitochondrial oxidative stress index between the patients and unaffected parental fibroblasts, corrected after NAC treatment [121]. Both patient cell lines exhibited reduced viability when grown in glucose-free medium. This was not improved by NAC in NDUFS8 mutant, while in contrast, NAC treatment fully restored cell viability in the ND5 mutant. 

Vitamin C or ascorbate: Acid ascorbic also ranks among the classical, widely used, antioxidant compounds. When treated with 10 μM ascorbate for 72 h the four aforementioned patient cell lines with translation-related proteins defects exhibited reduction of ROS levels and improvements in cell growth in glucose-free medium, for two of them [111]. Ascorbate induced little changes in ATP levels, mitochondrial content and membrane potential. In a previously discussed study [126], antioxidant effects of ascorbate on ROS levels were observed in three cell lines, while ROS tended to increase in four other ascorbate-treated cell lines. Regarding the cellular ATP content, both positive and negative effects of ascorbate were observed. 

Trolox: The data obtained with the antioxidant Trolox, a water-soluble vitamin E derivative, appeared more coherent [112,113,114]. In a first study, fibroblasts from six patients with mutations of various nuclear-encoded CI subunits were considered, in which CI protein amount and enzyme activity were generally reduced [114]. Higher ROS levels were measured in this panel of patient cells compared to control. Chronic treatment with 0.5mM Trolox for 96h decreased ROS levels in the six patient cell lines, four of them returning to the control range. Interestingly, in parallel with improvements in ROS levels, treatment with Trolox, induced significant increases in CI amount and enzyme activity, and therefore mitigated the CI deficiency. In a subsequent study, elevated ROS levels were found in 4 additional CI-deficient patient fibroblasts and, overall, in the panel of 10 patient fibroblasts considered in this study, a strong inverse correlation was revealed between ROS levels and mitochondrial membrane potential values (Δψ) [113]. Treatment with 0.3 mM Trolox for 72 h reduced ROS levels and restored Δψ. In addition to CI, the activities of RC Complex IV and of citrate synthase were also found upregulated, in control and in CI-deficient fibroblasts, in response to Trolox [112]. 

Resveratrol: It is widely admitted that RSV could exert part of its protective effects by virtue of its antioxidant properties. RSV may act by direct scavenging of ROS, given its polyphenolic structure, and/or by inducing anti-oxidative enzymes. Antioxidant effects of RSV in the context of inborn RC deficiency were investigated in several studies. In a series of six CI-deficient patient fibroblasts, treatment with 5 µM RSV for 72 h had no positive effects on ROS production and cell growth [110]. This could be likely explained by the low concentration (5 μM) used in this study. In a second study, the same group used 25 μM RSV for 72 h to treat four cell lines with mitochondrial translation defects. The authors reported little or no changes in ROS overproduction and sporadic positive effects of RSV treatment on ATP content or cell growth in some fibroblasts but not in others; overall, the effects of RSV were quite limited [111]. Finally, mostly variable results were also reported by the same group in 2017, with no clear-cut effects of 25 µM RSV on ROS levels, and on various parameters measured in RC-deficient patients [126]. Contribution from our laboratory has characterized the mechanism by which RSV could alleviate the excessive ROS production in some CI-deficient fibroblasts. Indeed and as aforementioned (Section 3.2) it was shown that treatment with 75 µM RSV stimulated residual CI enzyme activity, which resulted in the correction of RC deficiency in moderate CI-deficient patients’ fibroblasts [116]. We then rationally sought to probe for other possible beneficial effects of RSV on CI deficiency, and therefore assessed in parallel its antioxidant and metabolic effects in another panel of CI-deficient fibroblasts [117]. Eight out of 13 patient fibroblasts exhibited a significant stimulation of whole cell oxygen consumption rates in response to RSV treatment (75 μM, 72 h), and near normal levels were reached in five of them, illustrating marked improvements in RC deficiency. In these latter, ROS levels were much higher than normal prior to treatment, and were strongly reduced after exposure to RSV. Interestingly, significant RSV-induced increases in residual CI enzyme activity were also found, in parallel with the reduction in ROS levels. Treatment with RSV also increased the amount of the mitochondrial Mn-dependent superoxide dismutase 2 (SOD2) in the CI-deficient and the control cells, and this was no longer observed if cells were co-treated with the ER antagonist ICI182780, demonstrating the participation of ER in the RSV signaling cascade. Furthermore, the use of the inverse ERRα agonist XCT790 proved the involvement of ERRα in the induction of SOD2 by RSV in the patient cells. Unexpectedly, it was found that the increases in SOD2 protein were not accompanied by parallel increases in SOD2 enzyme activity in the patient cells, while increases were found in control cells. We sought to probe for the possible role of SIRT3 since this NAD^+^-dependent sirtuin is mandatory to deacetylate and activate SOD2 [223,224]. It could be hypothesized that, in the patient cells, the low NAD^+^ levels limited SIRT3 activity and hence resulted in reduced SOD2 activity. Interestingly, it was reported that while at basal level, cardiac mitochondria from SIRT3 -/- mice showed no changes in SOD activity and protein carbonylation levels compared to WT mice, SIRT3 -/- hearts subjected to ischemic–reperfusion injury exhibited increased mitochondrial ROS levels, higher mitochondrial protein carbonylation but lower SOD activity [225]. Therefore, the effects of RSV in the CI-deficient cells could not be attributed to the induction of cellular antioxidant enzymes, but rather to its direct antioxidant chemical properties. This was supported by the fact that treatment with a RSV tri-methoxy derivative failed to reverse ROS overproduction [117]. Taken together, these data confirmed that RSV is endowed by dual antioxidant properties, and clearly supported the notion that, in CI-deficient patient fibroblasts, treatment with RSV could cope with ROS overproduction and could upregulate CI residual enzyme activity, leading to alleviate, and eventually normalize, the respiratory chain defect. The hypothesized molecular mechanisms that could account for the effects of 75 µM RSV in human fibroblasts are schematized in Figure 13.

## 7. NAD^+^ Precursors

The data obtained with RSV regarding its antioxidant properties in CI-deficiency highlighted the pivotal role of NAD^+^, in keeping with the fact that the cellular and mitochondrial NAD^+^ to NADH ratio, and hence the homeostasis of these nucleotides, might be markedly affected as a result of inborn RC deficiencies. According to an old dogma, these pyridine nucleotides have been confined for decades to their role as oxido-reduction cofactors of many enzymes operating in intermediary and energy metabolism. A renewed interest has emerged when scientists discovered that NAD^+^-synthetizing and NAD^+^-consuming enzymes were in fact involved in many key cellular processes, strengthening the importance of this nucleotide in cell homeostasis and physiology [226,227,228]. Indeed, NAD^+^ is a co-substrate of several signaling enzymes among which appear the sirtuins and the poly(ADP-ribose)polymerases (PARP). The PARP protein superfamily comprises several enzymes, which modify target proteins post-translationally with ADP-ribose using NAD^+^ as substrate [229]. One of the best studied is PARP1, which senses and transduces signals to repair DNA lesions in cells, and which, upon activation, can deeply deplete the intracellular NAD^+^ levels [230]. Of note, PARPs are targets for cancer therapy, with several PARP inhibitors in preclinical development or in clinical use [229,231]. Indeed, PARP inhibitors are potent chemosensitizers and were shown to selectively inhibit the growth of cancer cells [231]. The discovery that alterations of NAD^+^ levels might occur with aging and with its associated chronic diseases, or in various pathological conditions, such as cardiac ischemia, obesity, type-2 diabetes, or neurodegenerative diseases [232,233,234], incited many scientists to seek ways to increase NAD^+^ pool at the cell or mitochondria level [235,236,237]. In theory, the replenishment of NAD^+^ levels can be obtained i) by stimulating NAD^+^ synthesis from its different precursors through supplementation with nicotinamide riboside (NR), nicotinamide mononucleotide (NMN) [238], nicotinic acid (NA), or nicotinamide (NAM); or ii) by inhibiting NAD^+^-consuming enzymes, in particular the PARPs. These assumptions have been tested with success in different mouse models of common diseases and, without going into detail, the resulting improvement in oxidative metabolism and mitochondrial biogenesis were generally attributed to the activation of the SIRT1/PGC-1α axis [239,240]. It can therefore be surmised that activation of the SIRT1/PGC-1α axis via manipulation of NAD^+^ levels could orchestrate the expression of transcriptional programs to restore mitochondrial functions in FAO or RC-deficient cells. In this regard, enhancement of NAD^+^ content might be beneficial at two levels, first by directly replenishing the depleted NAD^+^ pool, in CI disorders for instance, and second by providing NAD^+^ to sirtuins, in particular SIRT1. Along similar lines, sirtuin activation has recently been proposed as a promising therapeutic approach to treat inborn errors of metabolism (reviewed in [241]).

In patient fibroblasts, attempts to modulate NAD^+^/NADH ratio ex vivo were first performed using a PARP-1 inhibitor, MRL-45696, in CI-deficient fibroblasts from a patient with nuclear mutations in the NDUFS1 gene [120]. The authors reported that MRL-45696 efficiently enhanced the NAD^+^ content and increased in parallel the oxygen consumption rate, the β-oxidation of oleic acid and the citrate synthase activity, indicating an alleviation of RC deficiency and improved mitochondrial functions. Intriguingly, two other PARP-1 inhibitors (6-5H-phenanthridinone and PJ-34) were tested on the same NDUFS1-deficient fibroblasts by other authors who reported no increase in NAD^+^ content [118]. However, augmentations of mitochondrial membrane potential and of mitochondrial content were observed. Both inhibitors increased mRNAs of different mitochondrial-encoded RC subunits but not of nuclear-encoded ones. Finally, as an alternative strategy, supplementation with NR was tested, and this resulted in increased mitochondrial membrane potential and enhanced mitochondrial biogenesis in response to a seven-day treatment in the fibroblasts. Overall, the lack of increase of mitochondrial- and nuclear-encoded RC subunits in response to NR treatment, and the fact that NR increased NAD^+^ content, in contrast to PJ-34, suggested that the different strategies used to boost the NAD^+^ content did not trigger the same signaling pathways in human fibroblasts.

Several mouse models were used to investigate the in vivo effects of NAD^+^ precursors or PARP inhibitors on mitochondrial energy metabolism. The first study was performed in mice harboring cardiac-specific ablation of *Ndufs4* gene [242] resulting in altered assembly and/or stability of CI, and in decreased CI enzyme activity. Intriguingly, the heart-*Ndufs4* mice had normal lifespan and a near-normal cardiac function, except for an increased susceptibility to chronic cardiac stress. The CI deficiency was shown to result in 50% decrease of the NAD^+^/NADH ratio. It was surmised that this decrease in NAD^+^ level subsequently inhibited SIRT3 activity leading to an increased mitochondrial protein acetylation, responsible for the cardiac phenotype. The authors then sought to improve the NAD^+^ homeostasis by supplementing the heart-*Ndufs4* mice with NMN. This indeed translated into an increased NAD^+^/NADH ratio and a decrease of mitochondrial protein acetylation in heart, in the treated animals [242]. Another study was carried out in a *Nduf4* whole body knockout (KO) mice that presented several hallmarks of Leigh syndrome, in particular a progressive encephalopathy, in which the effects of a PARP-1 inhibitor, PJ-34 were studied [243]. The authors reported that, although the drug treatment did not prolong *Nduf4* mice survival, it improved the neuroscore and motor performance, which delayed the progression of the disease. Of note, it was reported that the PJ-34 inhibitor increased mRNAs of various mitochondrial and nuclear RC subunits, and the mtDNA content, in a tissue-specific manner, suggesting an improvement of mitochondrial functions. In line with these latter observations, a 72-h treatment with PJ-34 or with Olaparib, another potent PARP inhibitor, was found to increase by 25% the mitochondrial membrane potential in *Ndufs4* KO cultured glial cells. Finally, a subtle analysis of mitochondrial number and cristae area also indicated a remarkable positive effect of PJ-34 on these parameters in an organ-specific manner [243]. Overall, it can be concluded that the treatment with PJ-34 actually targeted the mitochondria and led to improve some features of mitochondrial bioenergetics. Several approaches to increase the NAD^+^ pool were also tested in another mouse model of mitochondrial disease, the *Sco2^KOKI^* mouse (Section 2.2) [244]. First, the authors created a double mouse mutant by crossing the *Sco2^KOKI^* mouse with a constitutive *Parp1-/-* mouse, which exhibited an elevated NAD^+^ level in the muscle. The suppression of *Parp1* in *Sco2^KOKI^* mouse improved the motor performance, which was accompanied by a significant increase of the otherwise reduced CII, CIII, and CIV enzyme activities. Next, the effects of NR administration were studied in *Sco2^KOKI^* mouse, and the authors found an increased NAD^+^/NADH ratio, in parallel with an improvement of motor performance and an induction of both nuclear- or mitochondrial-encoded OXPHOS genes in the skeletal muscle of *Sco2^KOKI^* receiving NR. However, no increase of mtDNA content or citrate synthase activity was reported upon NR treatment. The Tfam mRNAs levels were increased while no change was observed in PGC-1α transcripts. Of note, utilization of the PARP inhibitor MRLB-45696 improved both motor performance and mitochondrial function in *Sco2^KOKI^*, but did not increase NAD^+^/NADH ratio, or Tfam transcripts, despite an increased expression of mitochondrial-encoded genes. The idea that modulation of NAD^+^ level might be an effective therapy for mitochondrial disorders was further explored by Khan et al. [245]. Indeed, the authors showed that NR supplementation to Deletor mice, a good model of mitochondrial myopathy, triggered a mitochondrial biogenesis and hampered the development of mitochondrial ultrastructure abnormalities, which likely accounted for the delayed disease progression. Of note, increased levels of MCAD mRNA suggested stimulation of mitochondrial FAO pathway. Finally, more recently, Douiev et al. have reported the increased occurrence of nuclear double stranded DNA breaks (DSB) in COX-deficient fibroblasts suggesting a genomic instability. Excessive ROS production or reduced ATP did not seem to be major contributors to these nuclear DNA damages. Interestingly, the authors showed that NR treatment reduced DSB, and surmised the existence of NAD^+^ depletion as a cause of DSB in COX-deficient fibroblasts. In contrast, PARP inhibitor increased DSB [125]. Collectively, all the data summarized here bring evidence that the use of either NAD^+^ precursors or PARP inhibitors in various ex vivo or in vivo models of RC deficiency could result in improvements of mitochondrial functions. However, these data also highlight that the cascades of molecular events underpinning the beneficial effects of NAD^+^ precursors, or of PARP inhibitors, are not identical and, surprisingly, that these effects are not necessarily mediated through a modulation of NAD^+^ content. At present, and in light of these sometimes discrepant data, it is difficult to draw clear conclusions concerning the mode of action of these two classes of pharmacological molecules. Further studies are necessary, in particular in fibroblasts since only one cell line was tested, to decipher the precise molecular mechanisms underlying the effects of the various molecules tested. 

In conclusion, studies of the therapeutic potential of NAD^+^ precursors or PARP inhibitors appear promising in the field of RC disorders and should definitely be pursued, all the more so since Olaparib, a PARP inhibitor is currently marketed, and that both class of molecules have been, or are presently tested in several common diseases (ClinicalTrials.gouv). For instance, Trammelle et al. showed that a single NR oral administration of 100, 300, or 1000 mg to 12 healthy women and men, significantly increased blood NAD^+^ without serious adverse effects [246], and most relevant to mitochondrial disorders, two weeks of acipimox treatment, a nicotinic acid analog, was reported to significantly increase mitochondrial content and mitochondrial oxidative capacity in human skeletal muscle [247]. 

## 8. Protein Kinase A (PKA) Agonists

A potentially important cellular signaling pathway to manipulate mitochondrial oxidative metabolism is the PKA/cAMP signaling mechanism, all the more so since a large number of β-adrenoreceptor agonists are available, several of which being commonly used for the treatment of various pathologies [248,249]. It is now established that upon activation by β-adrenergic agonists, the PKA/cAMP system mediates the phosphorylation of mitochondrial RC complex I, IV, and V subunits, and of TCA enzymes, and these post-translational modifications play a major role in the functional regulation of mitochondrial oxidative metabolism [127,250,251,252,253,254]. For example, in human fibroblasts, exposure to 1 μM isoproterenol promotes within one hour the catalytic activity of CI, and lowers ROS [248,252]. HeLa cells treated for 1 h by 1mM 8Br-cAMP, a membrane permeant PKA agonist, exhibit a highly significant 25% increase in oxygen consumption [250]. Among 12 specific β2-adrenoreceptor agonists, which are approved drugs for various diseases, the group of Schnellmann identified five compounds that potently increased the maximal oxygen consumption rate in rabbit kidney proximal tubule cells after 24 h of treatment at 0.01 to 1 µM [249,253]. This β-adrenergic signaling cascade is physiologically active in the skeletal muscle and in other tissues under various stress conditions such as cold exposure or exercise, in which it mediates adaptive increases in mitochondrial FAO and oxidative metabolism [254,255].

In addition to mitochondrial proteins, the PKA-mediated cAMP signaling also induces the phosphorylation of CREB, which binds to DNA consensus sequences in the promoters of various nuclear target genes. PKA and the transcription factor CREB play a critical role in the biosynthesis of CI and CIV subunits [251]. In addition, the transcriptional activity of CREB is also responsible for various long-term metabolic effects documented in several cell lines and in animal studies, through the upregulation of downstream genes encoding PGC-1α, NRF1, NRF2, and ERR transcription factors, some FAO enzymes, and mtTFA [251,253,255]. Thus, evidence has emerged that the cAMP/PKA/CREB cascade can serve long-term adaptations of energy homeostasis via stimulation of the fatty acid oxidation and respiratory chain (OXPHOS) activity, and of the mitochondrial biogenesis. Furthermore, there are data showing that pharmacological activation of PKA has a protective role to combat mitochondrial ROS overproduction and oxidative damage to RC complex protein subunits [248,251,252], and OXPHOS protein phosphorylation might be one of the mechanisms by which cells adapt to bioenergetics defects and ROS generation [251]. The cellular cAMP-dependent signaling is highly complex and multifaceted, and is now admitted to involve different subcellular pools of cAMP and PKA. In particular, it is known that phosphorylation of RC complex subunits is mediated at least partly by intra-mitochondrial soluble adenylyl cyclase and PKA [250]. Overall, new regulations of mitochondrial functions by cAMP-PKA signaling are emerging, which could play a major role in modulating energy production and mitochondrial biogenesis in response to various stresses [254]. Yet, very few studies have explored the therapeutic potential of β-adrenergic receptors agonists in cells with genetic respiratory chain or FAO deficiencies, In fibroblasts from a patient with severe CI deficiency due to NDUFS1 gene mutations, it was reported that exposure to dibutyryl-cAMP (100 μM, 1 h) stimulated the residual CI activity, and induced disappearance of ROS [119]. In cultured cybrid cells harboring various mtDNA mutations and in fibroblasts from patients with mutations of the nuclear gene *Sco2*, responsible for COX deficiency, Acin-Perez et al. showed that exposure to 1 mM 8-Br-cAMP induced a marked stimulation of COX activity, in parallel with increases in cell respiration and ATP synthesis [127]. Finally and to our knowledge, no in vivo studies in mouse model of RC or FAO deficiency have been conducted to evaluate this class of compounds. Accordingly, screening the existing approved drugs acting as β-adrenergic agonists in RC- or FAO-deficient patient fibroblasts clearly warrant further investigation, and would undoubtedly be of great interest in the search of pharmacological agents to treat these disorders.

## 9. Other Molecules, Other Pathways

As previously mentioned, this review does not aim at covering all the approaches under study for the treatment of FAO or RC disorders, and in particular did not address the effects of dietary manipulations, which have garnered a considerable amount of attention as potential therapy in this class of disorders. Very briefly, it can be mentioned that nutritional supplements containing triheptanoin, an anaplerotic odd-chain fatty acid, have now been extensively tested in clinical trials in a variety of FAO disorders, and appear promising to improve some of the disease manifestations [256,257]. Implementation of a ketogenic diet has also been proposed in some patients with RC deficiencies [12]. Administration of riboflavin led to attenuate the muscular symptoms in patients with late-onset multiple acylCoA dehydrogenase deficiency (MADD), but the mechanisms accounting for these beneficial effects remain elusive [258,259,260]. 

Among the powerful effectors of energy metabolism, known for a long time, is mitochondrial calcium. Recent data pointed out the importance of endoplasmic reticulum (ER)-mitochondria tethering in regulating mitochondrial Ca^2+^ homeostasis, and allowed to identify several proteins participating to this inter-organelle contact sites called mitochondria-associated endoplasmic reticulum membranes (MAM) [261,262]. Ca^2+^ transfer from ER, driven by trans-membrane potential, is indispensable for preserving cell energetics, and the protein machinery involved in the mitochondrial Ca^2+^ influx and efflux has recently been elucidated [263,264]. Mild increases in mitochondrial Ca^2+^, acting via the local PKA signaling pathway, induce allosteric activation of several TCA cycle and OXPHOS enzymes, leading to a stimulation of mitochondrial respiration and ATP synthesis. Mitochondrial Ca^2+^ dynamics therefore plays a major role in controlling energy production, and unbalanced mitochondrial Ca^2+^ levels have been proposed as a pathogenic factor in several human diseases [263,264]. Accordingly, it is admitted that pharmacological modulators of mitochondrial Ca^2+^ homeostasis targeting the uptake or release pathways might have therapeutic potential for the treatment of these disorders. Nevertheless, the compounds actually known to influence mitochondrial Ca^2+^ balance (lanthanides, ruthenium red, benzothiazepines, verapramil, cyclosporine A) have major drawbacks due in particular to interactions with extra-mitochondrial targets, and new compounds are under study for pharmacological modulation of mitochondrial Ca^2+^ homeostasis [263,264].

Interestingly, recent studies revealed that interactions with other organelles also contribute to the regulation of mitochondrial metabolism. In particular, physical interactions between mitochondria and peroxisomes much likely contribute to the ß-oxidation of fatty acids, and both organelles are involved in ROS homeostasis [265]. Furthermore, peroxisomes and mitochondria share proteins of their division machinery, like Drp1, and tether proteins [266]. In a recent study, overexpression of the newly characterized Pex34 tether protein in yeast was shown to greatly increase the rate of CO2 production from octanoate, therefore demonstrating the functional importance of peroxisome-mitochondria contact sites in the cell β-oxidation capacity [267].

## 10. Conclusions

Overall, it is evident that several drugs or small molecules have the ability to improve or correct FAO- or RC-deficient bioenergetics functions in various models of inborn mitochondrial disorders, acting via different endogenous metabolic signaling pathways (Table 3). The use of cellular models can be particularly helpful to progress in the proposition of new pharmacological treatments and to bring insights into their mechanisms of action. A number of important considerations emerge from the data obtained after cell treatment with various compounds. For instance, it is clear that inborn defects in one single enzyme or protein in the FAO or RC pathways have multiple direct and indirect consequences on energy homeostasis, and trigger complex cellular adaptive mechanisms, and this raises the question of which outcome(s) should be seen as optimal to objectivize the effects of candidate molecules. In this regard, we found that functional studies of FAO or RC capacities in intact fibroblasts, based on measurements of tritiated LCFA or of oxygen consumption rates, provide robust quantitative indexes to evaluate the extent of the metabolic deficiency on the one hand, and the changes induced after treatment by molecules of interest, on the other hand. In parallel, a number of other parameters can be used to characterize the pharmacological responses. Thus, data from control fibroblasts indicate that treatment by bezafibrate, resveratrol, or AMPK activators, generally result in a coordinate induction of numerous FAO or RC enzymes and proteins, accounting for the concomitant increases of mitochondrial functions. Gene mutations, however, will frequently result in marked instability of the corresponding mutant protein, making eventually tricky the quantification of the deficient protein or enzyme activity, and the evaluation of its changes in response to treatments. Nevertheless, analysis of available data lends support to the notion that, in treated patients’ fibroblasts, the increase in mutant protein level or enzyme activity is a determining factor in the restoration of FAO or RC capacities. Conversely, it should be emphasized that pharmacological treatments targeting mitochondrial bioenergetics will most likely be ineffective if the disease-causing gene mutations fully abolish the synthesis or activity of the gene product. Consequently, the pharmacological responses can variably be affected as a function of patients’ gene mutations. This might orient toward personalized genotype-based therapy, a “next generation therapeutics” successfully initiated for the pharmacological treatment of cystic fibrosis, in coming years [268,269,270]. 

Altogether, cell-based screening of drugs or natural compounds can provide pre-clinical proof-of-concepts to identify compounds of interest, and to determine the responsiveness to the treatment of patients’ fibroblasts with various disorders and various genotypes. However, this approach does not shed light on possibly beneficial, toxic, or unwanted effects of these compounds in affected patients, which can only be established by in vivo studies.

Finally, it is becoming increasingly clear that multiple bioenergetics regulatory factors can be targeted to mediate beneficial effects on mitochondrial oxidative metabolism. Mitochondria-oriented therapies by small molecules is a rapidly expanding field covering rare and common diseases, which could pave the way to rational interventions in a variety of disorders. Beyond the identification of individual candidate molecules, it would be worthwhile to test them in association, which could reveal synergistic effects enhancing their therapeutic potential. It can also be thought that combining multiple therapeutic rationales, i.e., small molecules, but also dietary interventions (triheptanoin, ketogenic diets), supplementation by co-factors (NAD^+^ precursors, riboflavin, CoQ10) and protein-repair therapies (chaperones, correctors, potentiators) could open new opportunities for the treatment of mitochondrial disorders.

## Figures and Tables

**Figure 1 cells-08-00289-f001:**
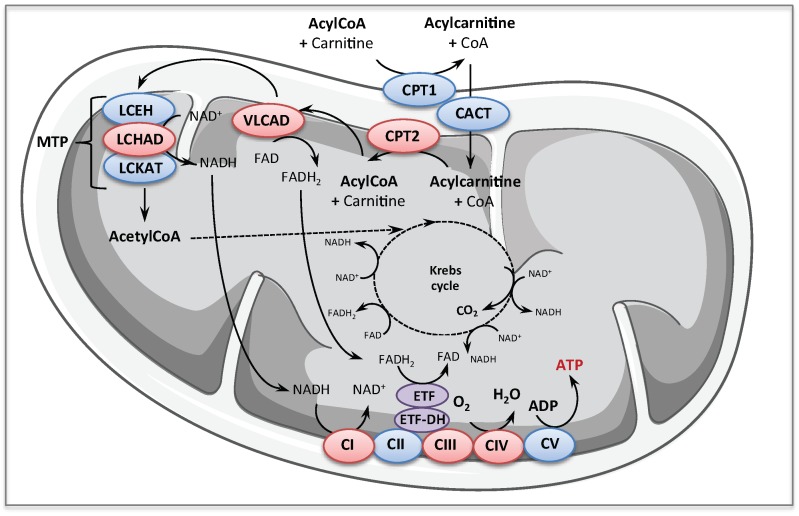
Schematic of the mitochondrial long-chain fatty acid oxidation and respiratory chain pathways. The enzymes mainly considered for pharmacological therapy in this review are in red: CPT1, carnitine palmitoyl transferase 1; CACT, carnitine acylcarnitine translocase; CPT2, carnitine palmitoyl transferase 2; VLCAD, very long chain acylCoA dehydrogenase; MTP, mitochondrial trifunctionnal protein; LCEH, long-chain enoyl-CoA hydratase; LCHAD, long-chain-3-hydroxyacyl-CoA dehydrogenase; LCKAT, long-chain 3-ketoacyl-CoA thiolase; ETF, electron transferring factor; ETF-DH, ETF dehydrogenase; CI, CII, CIII, CIV, and CV, Complex I, II, III, IV, and V of mitochondrial respiratory chain; NAD^+^, oxidized nicotinamide adenine dinucleotide; NADH, reduced nicotinamide adenine dinucleotide; FAD, oxidized flavin adenine dinucleotide; FADH2, reduced flavin adenine dinucleotide.

**Figure 2 cells-08-00289-f002:**
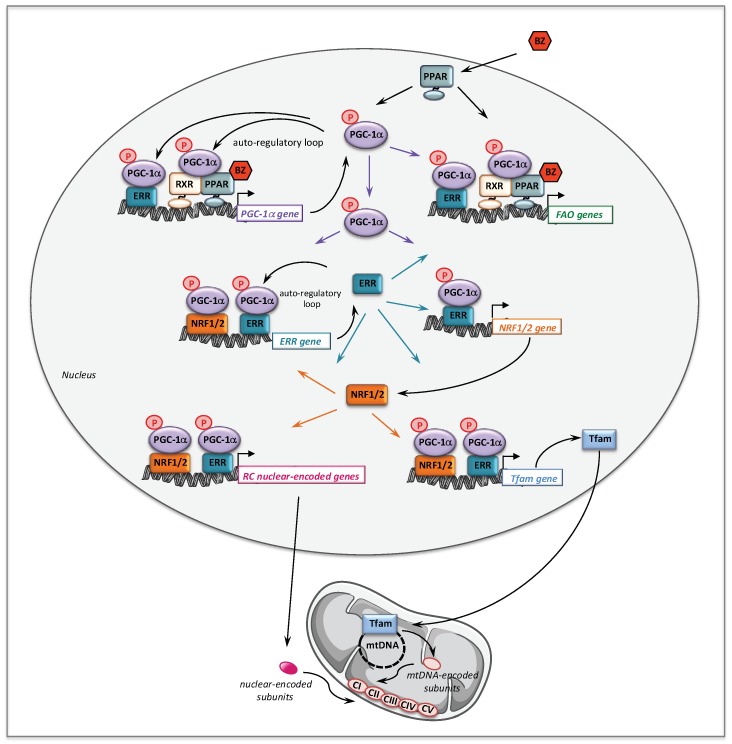
Molecular mechanisms underlying the effects of BZ on fatty acid oxidation and respiratory chain genes and on mitochondrial biogenesis. RXR: retinoid X receptor; PPAR: peroxisome proliferator activated receptor; PGC-1α: PPAR gamma co-activator 1 alpha; ERR: estrogen related receptor; NRF: nuclear respiratory factor; Tfam: mitochondrial transcription factor A.

**Figure 3 cells-08-00289-f003:**
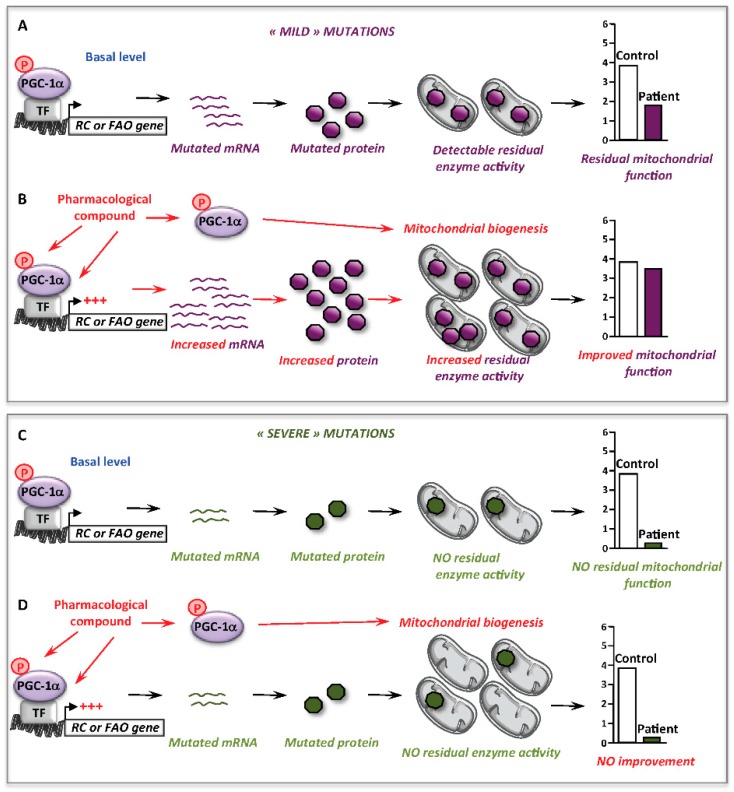
Involvement of mitochondrial biogenesis in the correction of mitochondrial RC or FAO deficiency. (**A**) At basal level, some ‘mild’ mutations allow the production of mutated proteins with some level of residual enzyme activity and mitochondrial function. (**B**) Upon pharmacological activation nuclear and mitochondrial gene transcription could be stimulated either directly or indirectly via PGC-1α, leading to increase the amount of mutated mRNAs and proteins, increase residual enzyme activity and improve or correct the deficient mitochondrial function. The activation of PGC-1α induces a mitochondrial biogenesis. (**C**) At basal level, ‘severe’ mutations often lead to unstable/undetectable mRNA and protein with no residual enzyme activity, which lead to severely affected mitochondrial function. (**D**) Upon pharmacological activation, although PGC-1α could induce a mitochondrial biogenesis, the bottleneck of the metabolic function linked to the mutations’ severity could not be released due to the absence of increased mutant proteins. PGC-1α: PPAR gamma co-activator 1 alpha; TF: transcription factor.

**Figure 4 cells-08-00289-f004:**
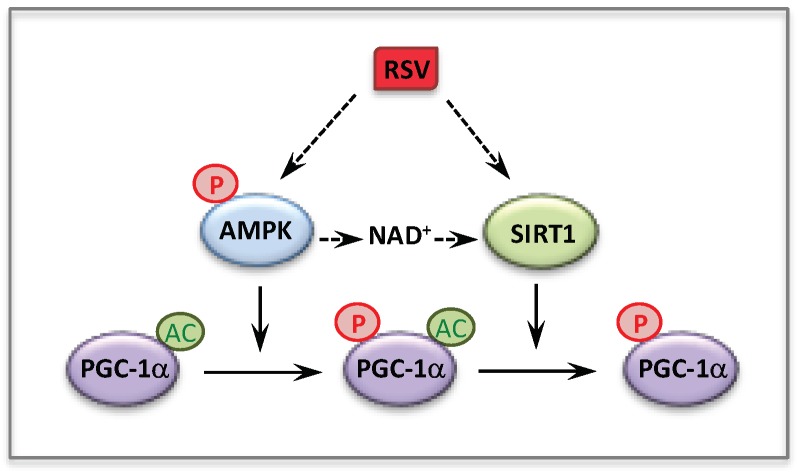
Activation of PGC-1α by RSV via AMPK and SIRT1.

**Figure 5 cells-08-00289-f005:**
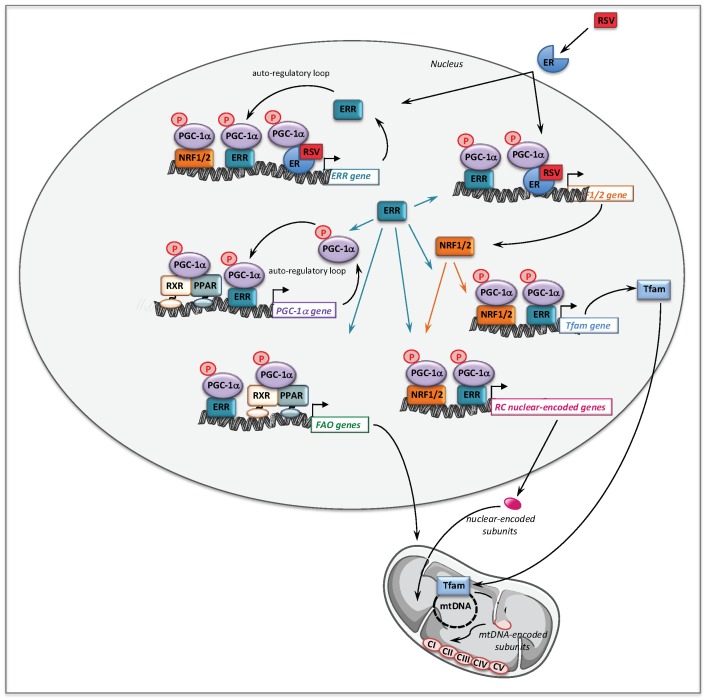
Molecular mechanisms underlying the effects of 75 µM RSV on fatty acid oxidation and respiratory chain genes in human fibroblasts. RXR: retinoid X receptor; PPAR: peroxisome proliferator activated receptor; PGC-1α: PPAR gamma co-activator 1 alpha; ER: estrogen receptor; ERR: estrogen related receptor; NRF: nuclear respiratory factor; Tfam: mitochondrial transcription factor A.

**Figure 6 cells-08-00289-f006:**
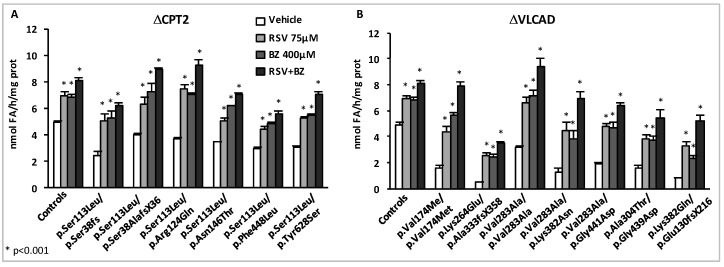
Effects of maximal doses of RSV and BZ in CPT2- or VLCAD-deficient fibroblasts with various genotypes [167]. FAO was measured by quantitating the production of ^3^H_2_O from (9,10-^3^H) palmitate as described previously [78]. For each cell line, FAO measurements were performed in triplicate, and repeated in at least three independent experiments.

**Figure 7 cells-08-00289-f007:**
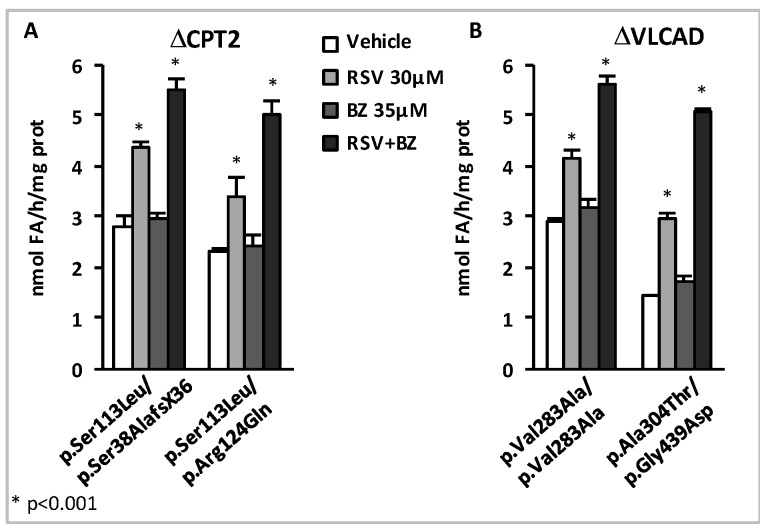
Synergistic effects of low concentrations of RSV and BZ [167]. FAO measurements were performed as described in Figure 6.

**Figure 8 cells-08-00289-f008:**
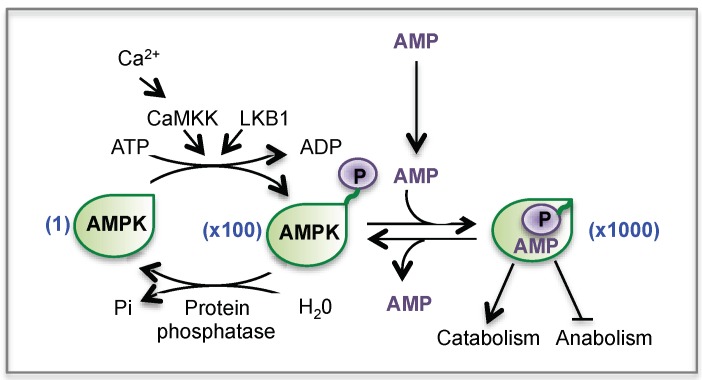
Regulation of AMPK by AMP and by phosphorylation. CaMKK: Ca2+/calmodulin-dependent protein kinase. LKB1: Liver kinase B1.

**Figure 9 cells-08-00289-f009:**
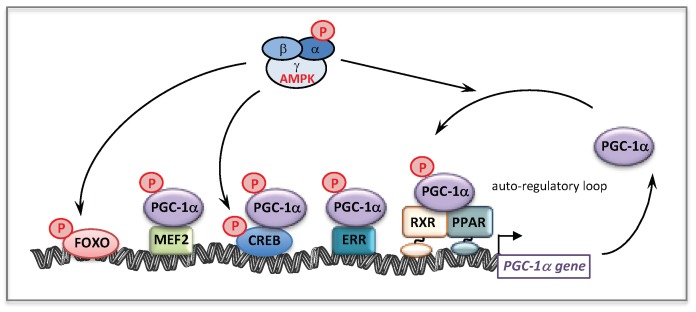
Effects of AMPK on transcription factors and on PGC-1α.

**Figure 10 cells-08-00289-f010:**
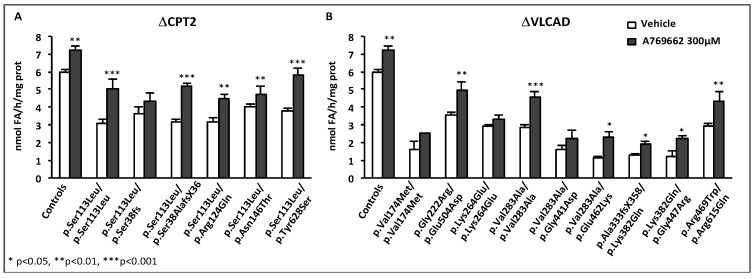
Effects of A769662 on fatty acid oxidation in CPT2 or VLCAD-deficient fibroblasts. FAO measurements were performed as described in Figure 6.

**Figure 11 cells-08-00289-f011:**
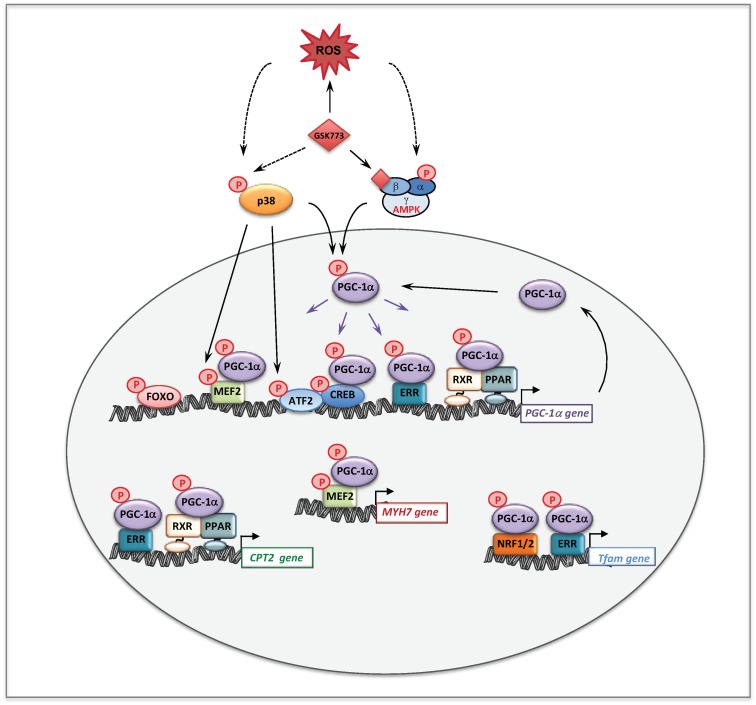
Proposed scheme for GSK773 molecular mechanisms. GSK773 is a direct and specific activator of AMPK. Its binding to the β1 subunit phosphorylates and activates AMPK. GSK773 induces p38 MAPK phosphorylation. GSK773 also induces an increase of ROS levels, which might participate to AMPK and p38 phosphorylation. Like AMPK (cf Figure 9), p38 MAPK can also directly phosphorylate PGC-1α, and several transcription factors such as MEF2 and ATF2, whose binding sites are present on PGC-1α promoter. These mechanisms might account for the GSK773-induced increase of PGC-1α transcription and protein levels. PGC-1α might then upregulate the transcription of CPT2, Tfam and MYH7, which encodes myosin heavy chain I (MHCI) expressed in oxidative slow twitch fibers. Finally, the coordinated increase of PGC-1α, Tfam, FAO proteins and certainly nuclear and mitochondrial encoded RC proteins are consistent with the mitochondrial biogenesis induced by GSK773.

**Figure 12 cells-08-00289-f012:**
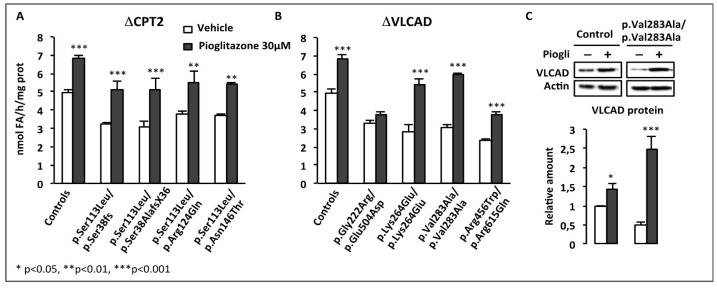
Effects of Pioglitazone on fatty acid oxidation in CPT2 or VLCAD-deficient fibroblasts. FAO measurements were performed as described in Figure 6 and western-blots as described in [87].

**Figure 13 cells-08-00289-f013:**
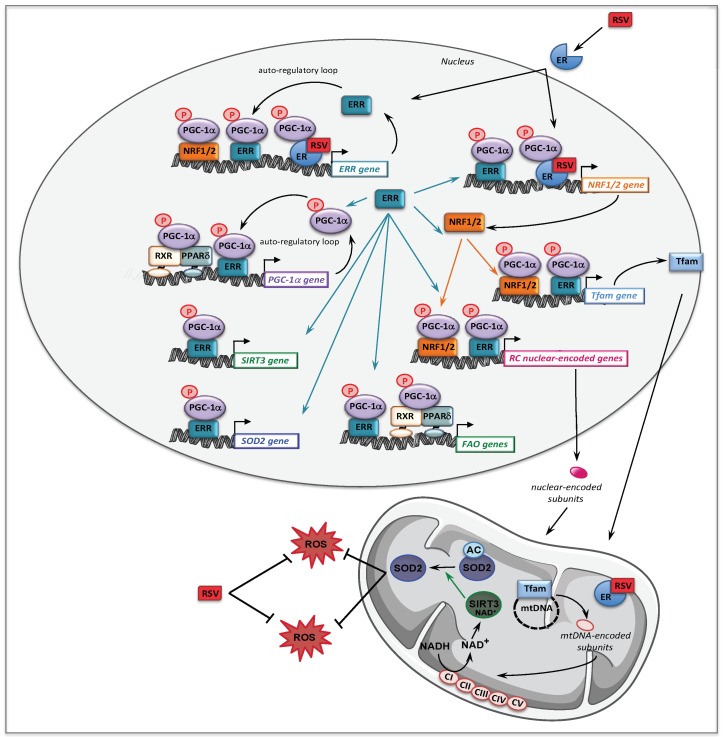
Hypothesized signaling pathways underlying the positive effects of RSV on mitochondrial metabolism and oxidative stress.

**Table 1 cells-08-00289-t001:** Genotypes of FAO-deficient patients for which pharmacological compounds have been tested.

Mutated Gene	Amino Acid Change	Amino Acid Change	Effects	References
	p.Pro50His	p.Asp608His	+	[97]
	p.Arg51Gly	p.Glu174Lys	+	[84]
	p.Ser113Leu	p.Ser113Leu	+	[78,80,97,98], this review
	p.Ser113Leu	p.Ser38fs	+	[78,99,100], this review
	p.Ser113Leu	p.Ser38AlafsX35	+	[97,98]
	p.Ser113Leu	p.Ser38AlafsX36	+	[100], this review
**CPT2**	p.Ser113Leu	p.Arg124Gln	+	[78,99,100], this review
	p.Ser113Leu	p.Asn146Thr	+	[78,100], this review
	p.Ser113Leu	p.Phe188Ser	+	[80]
	p.Ser113Leu	p.Trp201X	+	[97,98]
	p.Ser113Leu	p.Gly377Asp	+	[97,98]
	p.Ser113Leu	p.Phe448Leu	+	this review
	p.Ser113Leu	p.Tyr628Ser	+	[80,100]
	p.Asp328Gly	p.Asp328Gly	-	[78,100]
	p.Gln13X	p.Arg286Gly	-	[39,87]
	p.Gly34fsX60	p.Gly34fsX60	-	[39]
	p.Pro65Leu+Lys247Gln	p.Pro65Leu+Lys247Gln	-	[39]
	p.Pro89Ser	p.Ala536fsX550	±	[39]
	p.Pro91Gln	p.Gly193Arg	+	[39]
	p.Asn122Asp	p.Asn122Asp	-	[39,86,87]
	p.Ser148Gly	p.Arg511Trp	±	[39]
	p.Val174Met	p.Val174Met	+	[39,86,87,100], this review
	p.Val174Met	p.Glu609Lys	+	[101]
	p.Gly185Ser	p.Asn252_His293del42	-	[39,87,100]
	p.Gly185Ser	p.Gly294Glu	-	[39]
	p.Gly222Arg	p.Gly222Arg	-	[39,87]
	p.Gly222Arg	p.Glu504Asp	+	[39,99,100], this review
	p.Arg229X	p.Arg613Trp	-	[39]
	p.Asn252_His293del42	p.Gly441Asp	-	[39,87,100]
	p.Thr260Met	p.Ala640fsX679	-	[39,87]
	p.Lys264Glu	p.Lys264Glu	+; ±	[39,100], this review
	p.Lys264Glu	p.Ala333fsX358	±	[39,84], this review
**VLCAD**	p.Lys264Glu	p.Met437Val	+	[39,100]
	p.Val283Ala	p.Val283Ala	+	[39,86,99,100], this review
	p.Val283Ala	p.Lys382Asn	±	this review
	p.Val283Ala	p.Gly441Asp	+	[39,100], this review
	p.Val283Ala	p.Glu462Lys	±	[39,100], this review
	p.Val283Ala	p.?	+	[101]
	p.Lys299Met	p.Leu502Gln	±	[39]
	p.Ala304Thr	p.Gly439Asp	±	[39,87], this review
	p.Ala333fsX358	p.Lys382Gln	+	[39,100], this review
	p.Arg366His	p.Arg453X	±	[39,87]
	p.Lys382Gln	p.Gly447Arg	+; ±	[39,84,99,100], this review
	p.Lys382Gln	p.Glu130fsX216	±	[39,87], this review
	p.Asp405His	p.Arg450His	±	[39,87]
	p.Ala416Thr	p.Arg450His	±	[39,100],
	p.Ala416Thr	p.Lys600fsX679	±	[39,87]
	p.Arg453Gln	p.Arg453Gln	-	[39,87]
	p.Arg456His	p.Arg615Gln	+	[39,100], this review
	p.Arg469Trp	p.Arg469Trp	-	[39]
	p.Lys540Pro	p.?	+	[101]
	p.Thr37SerfsX6	p.?	+	[89]
	p.Lys267SerfsX7	p.Lys267SerfsX7	-	[89]
	p.Lys353IlefsX19	p.Lys353IlefsX19	-	[89]
	p.Glu446Lys	p.Gly703Arg	-	[89]
	p.Pro467_Ile495del	Pro467_Ile495del	+	[89]
	p.Glu510Gln	p.Glu510Gln	-	[89]
**HADHA**	p.Glu510Gln	p.Thr37SerfsX6	+	[89]
	p.Glu510Gln	p.Ile305Asn	+	[89]
	p.Glu510Gln	p.Gly328Arg and p.Gln358Lys	+	[89]
	p.Glu510Gln	p.Leu571Pro	-	[89]
	p.Glu510Gln	p.Arg676His	-	[89]
	p.Glu510Gln	p.?	-	[102]
	p.Lys664ValfsX2	p.Lys664ValfsX2	-	[89]
	p.Val705Asp	p.Val705Asp	-	[89]
	p.Asn114Ser	p.?	+	[89]
	p.Arg247Cys	p.Asp273IlefsX20	-	[89]
**HADHB**	p.Ser383Leu	p.Ser383Leu	-	[89]
	p.Asn389Asp	p.Asn389Asp	-	[89]
	p.Arg444Lys	p.Arg444Lys	-	[89]
	p.Val455Gly	p.Val455Gly	-	[89]
	p.Arg83Cys	p.Arg83Cys	±	[102]
**SCAD**	p.Arg83Cys	p.Gly185Ser	±	[102]
	p.Gly185Ser	p.Gly185Ser	±	[102]
	p.Arg380Trp	p.Arg380Trp	±	[102]
**MCAD**	p.Lys329Glu	p.?	±	[102]

+: positive effects; -: no or negative effects; ±: mixed effects.

**Table 2 cells-08-00289-t002:** Genotypes of RC-deficient patients for which pharmacological compounds have been tested.

Patients	Mutated Gene	Amino Acid Change	Amino Acid Change	Effects	References
		p.Arg59X	p.Thr423Met	+	[112,113,114]
		p.Tyr204Cys	p.Cys206Gly	+	[105,115,116]
		p.Glu214Lys	ex 8 del	+	[116]
	**NDUFV1**	p.Ala432Pro	p.Gly388X	+	[116]
		p.Glu377Lys	p.Glu377Lys	-	[116]
		p.Arg386His	p.Pro252Arg	-	[116]
		p.Arg386Cys	p.Ser251fsX44	-	[116]
		p.Gln381Arg	p.Arg386His	-	[117]
		p.Arg386Cys	p.Arg386Cys	+	[117]
	**NDUFV2**	ex 2 del	ex 2 del	+	[105,116]
		p.Ala183Thr	p.Tyr70fs*6	+	[117]
		p.Gly19_Val40del	p.Gly19_Val40del	+	[117]
	**NDUFS1**	Del222	p.Asp252Gly	-	[105]
		del entire gene	p.Met707Val	-	[116]
		P.Arg241Trp	p.Arg557X	-	[116]
		p.Asp380Val	?	-	[117]
		p.Val228Ala	p.Asp252Gly	±	[117]
**CI-deficient**		p.Gln522Lys	p.Gln522Lys	+	[118,119,120]
		p.Arg557X	p.Asp618Asn	-	[112,113,114]
	**NDUFS2**	p.Arg228Gln	p.Arg228Gln	+	[110,112,113,114]
		p.Met292Thr	p.Met443Lys	+	[117]
		p.Met292Thr	p.Arg118Gln	-	[117]
		p.Ser413Pro	p.Ser413Pro	-	[112,117]
	**NDUFS3**	p.Thr145Ile	p.Arg199Trp	+	[105,116]
	**NDUFS4**	ex 3-4 del	ex 3-4 del	-	[105]
		p.Trp97X	p.Trp97X	+	[112,117,119]
		p.Arg106X	p.Arg106X	+	[112,113]
	**NDUFS6**	ex 3 and 4 del	ex 3 and 4 del	-	[116]
		p.Leu23Trpfs*35	p.Leu23Trpfs*35	±	[117]
	**NDUFS7**	Ala6_Arg213del	Ala6_Arg213del	-	[117]
		p.Val122Met	p.Val122Met	+	[112,113,114]
		p.Arg145His	p.Arg145His	+	[117]
	**NDUFS8**	p.Arg54Trp	p.Gly20Arg	-	[121]
		p.Arg94Cys	p.Arg94Cys	±	[112,113,114]
	**C8orf38**	p.Gln99Arg	p.Gln99Arg	-	[116]
	**C20orfF7**	p.Gly250Val	p.Gly250Val	±	[110]
	**NDUFAF4**	p.Thr194Cys	p.Thr194Cys	±	[110]
	**FOXRED1**	p.Arg352Trp	p.Arg352Trp	±	[110]
	**NDUFA12L**	p.Met1Leu	p.Met1Leu	±	[110]
	**ACAD9**	p.Arg518His	p.Arg518His	+	[115]
**CII-deficient**	**NFU1**	p.Gly208Cys	p.Gly208Cys	-	[122]
	**FP**	p.Gly555Glu	p.Gly555Glu	-	[122]
**CIII-deficient**	**BCS1**	p.Pro99Leu	p.Pro99Leu	+	[105]
	**COX10**	p.Asn204Lys	p.Asn204Lys	+	[105,116]
	**SURF1**	p.Leu105X	p.Leu105X	-	[116,122]
		p.Gly180Glu	?	-	[116]
**CIV-deficient**		p.Pro183fsX189	p.Pro183fsX189	-	[105]
		p.Ser282Cysfs	p.Ser282Cysfs	-	[122]
	**LRPPRC**	P.Ala354Val	P.Ala354Val	-	[123]
	**COX6B1**	p.Arg20Cys	p.Arg20Cys	±	[124,125,126]
	**COX4I1**	p.Lys101Asn	p.Lys101Asn	+	[125]
	**SCO2**	p.Gln53X	p.Glu140Lys	+	[127]
	**EFT**	p.Arg333trp	p.Arg333trp	±	[111,126]
**Mutiple RC defects**	**GFM1**	p.Leu398Pro	p.Leu398Pro	±	[111]
	**MRPSS22**	p.Arg170His	p.Arg170His	±	[111,126]
	**TRMU**	p.Tyr77His	p.Tyr77His	±	[111]

+: positive effects; -: no or negative effects; ±: mixed effects.

**Table 3 cells-08-00289-t003:** Molecules tested in preclinical models of FAO and RC disorders.

Compounds		Disorders	References
	BZ (PPARα/δ)	FAO	[39,78,80,84,85,86,87,88,89,97,102]
		RC	[105,107,109,110,111,124,129,130,131,134]
	GW7647 (PPARα)	FAO	[80]
PPAR agonists		RC	[105]
	GW0742 (PPARδ)	FAO	[80]
		RC	[105]
	Pioglitazone (PPARγ)	FAO	this review
Stilbenes	trans-RSV	FAO	[99,100,101]
		RC	[110,111,115,116,117,122,124,126,149,150]
	cis-RSV	FAO	[99]
	piceid	FAO	[99]
RSV metabolites	RSV-3-O-glucuronide		[99]
	RSV-4-O-glucuronide	FAO	[99]
	RSV-3-O-sulfate		[99]
	di-hydro-RSV		[99]
AMPK activators	AICA-riboside	FAO	[100], this review
		RC	[110,111,124,129,192]
	A769662	FAO	[100], this review
	GSK773	FAO	[98]
Antioxidants	NAC	FAO	[101,102]
		RC	[111,115,124,126]
	Trolox	FAO	[101]
		RC	[112,113,114,115]
	MitoQ	FAO	[101]
		RC	[115]
	Quinone analogues	RC	[220,222]
	JP4-039	FAO	[101]
		RC	[115]
	XJB-5-131	FAO	[101]
	Vitamin C or Ascorbate	FAO	[102]
		RC	[111,126]
NAD^+^	NAD^+^ precursors	RC	[118,125,242,244,245]
	PARP-1 inhibitor	RC	[118,243,244]
PKA agonists	Dibutyryl-cAMP	RC	[119]
	8-Br-cAMP	RC	[127]

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
