# Peer review of "Mitochondrial Genetic Disorders: Cell Signaling and Pharmacological Therapies"

_cells, 2019, doi:10.3390/cells8040289_

Round 1

Reviewer 1 Report

This is a very rich and detailed review on the disorders in the mitochondrial energy and lipid metabolism and potential signaling and therapeutic interventions. Moreover, the review includes new experimental findings that support the current concept and foster the review in some parts. The review also includes multiple excellent schemas that are very well drawn and very clear. These are included always to summarize a subchapter and are very helpful for the reader. The review consists on various sub-chapters that handle e.g. the lipid signaling of mitochondria or the regulation of OXPHOS. Much appreciated is the efforts the authors accepted to provide some information the already possible therapeutic intervention. From this referee’s point of view, is the is review a very worthy piece of work that deserves further editorial consideration while there are some few aspects that should be addressed in a minor revision:

1.     Perhaps a Table that summarizes disorders and their molecular basis would be helpful for the readers

2.     While it is not necessary to be included in this work in extension, the recent finding son an impact on inter-organellar tethering, membrane potential and calcium homeostasis should be mentioned as further determinants for mitochondrial energy regulation.

3.     While the authors did an excellent job, this topic is huge and cannot be covered in one piece of review. Thus, the authors have selected what was most familiar/important for their opinion. While this is very fine, the should indicate and explain their selection and shortly mention the partial incompleteness on other aspects.

Minor issues:

-       Figures: At least on my copy there where several “?” instead of superscript numbers etc. Please check this on the original and the copy received by the journal.

-       Figure 1: the Krebs cycle is incomplete in terms of energy substrates (e.g. FADH2 is missing) and should be at least completed by all the electron carriers.

Author Response

Response to reviewer #1

First, the authors would like to thank the referees for their interesting and constructive remarks. We have taken them into account and changed the manuscript accordingly, as much as possible. We provide below answers to all the issues raised by the referees. We hope to have now improved the content and presentation of our review. Below are the details concerning all the changes brought to the manuscript, highlighted in grey.

We appreciate the fact that the reviewer acknowledged the addition of new experimental findings, which we want to share with the scientific community.

1.     Perhaps a Table that summarizes disorders and their molecular basis would be helpful for the readers

The revised manuscript contains modified Table 1 and Table 2, which summarize all the mutations/ genotypes of FAO- or RC- deficient patients that were addressed in our manuscript with the corresponding literature references. We hope that this answers the referee's request.

2- While it is not necessary to be included in this work in extension, the recent finding son an impact on inter-organellar tethering, membrane potential and calcium homeostasis should be mentioned as further determinants for mitochondrial energy regulation.

We thank the referee for the reminder of this indeed very interesting point, which we decided to discuss in the review page 37, 2d paragraph.

3- While the authors did an excellent job, this topic is huge and cannot be covered in one piece of review. Thus, the authors have selected what was most familiar/important for their opinion. While this is very fine, they should indicate and explain their selection and shortly mention the partial incompleteness on other aspects.

We agree with the referee that our explanations on this issue were too short. Accordingly, we have added mentions for the reader, both in the introduction page 2, 2d paragraph and in a new short paragraph pages 36-37, explaining our choices concerning the therapies of FAO or RC disorders discussed in our review. We hope that this important point will appear more clearly to the reader.

Minor issues

* Figures: At least on my copy there where several “?” instead of superscript numbers etc. Please check this on the original and the copy received by the journal.

We checked this point with the journal, and it appears that the original figures, sent and received, were fine, without any "?". They did not understand what the problem could be, and think that it might have something to do with the downloading process, which could have introduced these misprints. We hope that the problem will not happen again

* Figure 1: the Krebs cycle is incomplete in terms of energy substrates (e.g. FADH2 is missing) and should be at least completed by all the electron carriers.

The Krebs cycle in figure 1 is unfortunately too small to be completed with all the intermediates, but we have added the NAD/NADH and FAD/FADH2.

Reviewer 2 Report

Major comments:

1. Since 1) mitochondrial genetic diseases are associated with abnormalities in energy metabolism of mitochondria, and 2) the MS discusses genetic diseases, the title “Mitochondrial Genetic Disorders: Pharmacological Therapies” is more relevant.

2. Since this is a review paper, only results of previously published studies can be discussed in it. The authors discuss their new previously unpublished data (e.g., p.19, section 4, 1st paragraph and several places through the text). These data can be discussed in an original research paper in context with other parameters, detailed methods, etc. Therefore, Figs 6, 7, 10, 12, and 13 should be omitted. The authors should indicate the source (Ref.) of the Figure(s) in figure legends and obtain written consent from the journal(s) where the figure(s) has been published [if they were taken from previously published papers].

3. The MS can be improved by addition detailed characterization of mitochondrial metabolism and function for mitochondrial genetic disorders. For instance, in p.13 (1st paragraph) authors discuss Taz KO mice. In addition to ETC supercomplexes, Taz KO hearts demonstrated high basal but less Ca2+-induced mitochondrial swelling and ROS generation associated with high levels of cyclophilin D, a major regulator of mitochondrial permeability transition pore opening (Jang et al., 2017; PMID: 27604998).

4. Diseases associated with genetic deficiency of ETC individual complexes should be discussed in “1.2. Respiratory chain deficiencies” (e.g., Ugalde et al, 2004, PMID: 14749350; Koopman et al, 2007, PMID: 17428841; Antonichka et al, 2003, PMID: 12941961).

5. On p.32, 1st paragraph (line 15 from top): After “…and hence resulted in reduced SOD2 activity.” Should be indicated that “Interestingly, cardiac mitochondria from SIRT3 KO mice no changes in SOD activity and protein carbonylation levels (Parodi-Rullán RM et al, PMID: 28559847).

6. A table containing information about all pharmacological compounds currently used for targeting of mitochondrial energy disorders in preclinical studies and clinical trials should be involved in the review paper.

Minor comments:

Abbreviations should be given only once at first meeting in the text. 

Author Response

Response to reviewer #2

First, the authors would like to thank the referees for their interesting and constructive remarks. We have taken them into account and changed the manuscript accordingly, as much as possible. We provide below answers to all the issues raised by the referees. We hope to have now improved the content and presentation of our review. Below are the details concerning all the changes brought to the manuscript, highlighted in grey.

1- Since 1) mitochondrial genetic diseases are associated with abnormalities in energy metabolism of mitochondria, and 2) the MS discusses genetic diseases, the title “Mitochondrial Genetic Disorders: Pharmacological Therapies” is more relevant.

We have modified the title by including the term "genetic" as suggested by the referee.

2- Since this is a review paper, only results of previously published studies can be discussed in it. The authors discuss their new previously unpublished data (e.g., p.19, section 4, 1st paragraph and several places through the text). These data can be discussed in an original research paper in context with other parameters, detailed methods, etc. Therefore, Figs 6, 7, 10, 12, and 13 should be omitted. The authors should indicate the source (Ref.) of the Figure(s) in figure legends and obtain written consent from the journal(s) where the figure(s) has been published [if they were taken from previously published papers].

We agree with the referee that inclusion of some unpublished data may seem unusual for a review article. Hopefully, this only intended to complete the overview of numerous published results by providing a limited amount of original data that appeared directly relevant to some of the topics covered in the review.

- The results depicted in figure 6 and 7, dealing with the synergistic effects of bezafibrate and resveratrol, have been published as part of a patent filed by our team (ref 165 in the revised manuscript). We acknowledge that this was not clear enough in the submitted manuscript. The reference is now mentionned  in the legends of these figures.

- The figure 10 and 12, which bring new elements on the effects of TZD and of AMPK agonists fit with the context of the review, however we think that their content is, by itself, most likely too limited to form the body of a manuscript, and will thus never been published elsewhere. This is why we thought that including them in the review could be worthwhile and informative to the readers, without affecting the objective of the manuscript, i.e. to provide a global view based on analysis of a large amount of published literature.

It is essential to point out that these data were gathered using validated methods widely used in our laboratory, and respond to the same quality criteria as those used in our previous referenced publications. Thus, all the measurements of tritiated palmitate oxidation rates in patient cells after treatment by various small molecules were performed in triplicate and repeated at least on three independent fibroblasts cultures for each cell line, as now mentioned in the figures' legends.

- The figure 13 has been removed as requested by the reviewer,

3- The MS can be improved by addition detailed characterization of mitochondrial metabolism and function for mitochondrial genetic disorders. For instance, in p.13 (1st paragraph) authors discuss Taz KO mice. In addition to ETC supercomplexes, Taz KO hearts demonstrated high basal but less Ca2+-induced mitochondrial swelling and ROS generation associated with high levels of cyclophilin D, a major regulator of mitochondrial permeability transition pore opening (Jang et al., 2017; PMID: 27604998).

We understand the reviewer's comment who felt that not enough details were given in our submitted manuscript concerning the characterization of mitochondrial metabolism and functions in mitochondrial disorders. Accordingly, in the paragraph concerning “bezafibrate and respiratory chain disorders”, we have added a new paragraph regarding the published data on mitochondrial functions in animal models (pages 13-14). Concerning the Taz KO mice, we have re-written the paragraph and included the reference suggested by the reviewer, which is indeed very interesting (page 14).

4- Diseases associated with genetic deficiency of ETC individual complexes should be discussed in “1.2. Respiratory chain deficiencies” (e.g., Ugalde et al, 2004, PMID: 14749350; Koopman et al, 2007, PMID: 17428841; Antonichka et al, 2003, PMID: 12941961).

We agree with the reviewer that the paragraph “1.2 Respiratory chain deficiencies” did not provide enough information on the presentation of RC deficiencies and was too short compared to the one on FAO disorders. We have therefore added new paragraphs (pages 4-6), with the suggested references, on the biochemistry, molecular basis, and clinical features of the RC CI and CIV deficiencies, which are considered as the most frequent mitochondrial disorders.

5- On p.32, 1st paragraph (line 15 from top): After “…and hence resulted in reduced SOD2 activity.” Should be indicated that “Interestingly, cardiac mitochondria from SIRT3 KO mice no changes in SOD activity and protein carbonylation levels (Parodi-Rullán RM et al, PMID: 28559847).

We have added a mention to this interesting article, which had escaped our bibliography monitoring (page 32).

6- A table containing information about all pharmacological compounds currently used for targeting of mitochondrial energy disorders in preclinical studies and clinical trials should be involved in the review paper.

A new table (Table 3) summarizing the pharmacological compounds tested in the preclinical models discussed in the review has been added and will certainly improve the manuscript. However, and as mentioned in our review, only a brief analysis of some clinical trials was included, but we did not intend to review the results obtained in all clinical trials with all the small molecules tested in FAO or RC-deficiencies. Indeed, we certainly agree that this is an exciting topic, covered by many excellent reviews cited in our text, but which goes beyond the scope of our review. Adding a table on this issue would necessarily imply to analyze a large number of additional publications, and would require appropriate comments in the body of the text i.e. would likely deserve a whole manuscript, by itself.

- Minor comments:

Abbreviations should be given only once at first meeting in the text. 

We have checked that abbreviations were only given once along the text.

Reviewer 3 Report

This is a very extensive review; summarizing attempts validate various small molecules as therapeutic options for mitochondrial disorders caused by mutations in genes encoding proteins (mostly enzymes) involved in fatty acid oxidation (FAO) and the respiratory chain (RC).  Additionally new data are added.  

This is a very complicated subject as many diseases and molecules are discussed involving many different pathways.  Nevertheless the authors have succeeded to give a comprehensive overview and the figures and tables are certainly helpful.  

I have only minor comment as follows.

1. Mention that FAO contributes electrons to RC via NADH and CoQ

1. Mention that phenotypes found in muscle , sometimes is not expressed in fibroblasts

       1.1. Mention Acylcarnitines on blood as a diagnostic tool for FAOX

       2.2.  Re  partial correction of  FAO flux with Bezafibrate (and other molecules);  Please relate-if corrected to 50% this would  actually be sufficient, since carriers have approximately ~ 50% activity and they are generally healthy

2.2. Re patient trials; were the CPT2 VLCAD patient mutations known? Any correlation to fibroblasts results , carrying the same mutations?  

3. Consider mentioning the “French paradox”  

6.2. Consider that complexes I and III are generally thought to be the major sites of  ROS production and its questionable if  defect in other complexes  would produce ROS      

6.2. Consider discussing data from PMID: 28093355

7. Mention that  PARP inhibitors are sometimes used in cancer treatment

7. Consider discussing data from PMID: 29886046

Tables 1 and 3: It would be helpful to add another column- summarizing compounds and effects:  positive/negative/mixed (some parameters positive some negative) /not significant

All figures ; There seem to be question marks   in the text boxes???.  Is this due to conversion to pdf??

Author Response

Response to reviewer #3

First, the authors would like to thank the referees for their interesting and constructive remarks. We have taken them into account and changed the manuscript accordingly, as much as possible. We provide below answers to all the issues raised by the referees. We hope to have now improved the content and presentation of our review. Below are the details concerning all the changes brought to the manuscript, highlighted in grey.

1. Mention that FAO contributes electrons to RC via NADH and CoQ

 We have mentioned that FAO contributes electrons to RC via NADH and FADH2 (page 2).

Mention that phenotypes found in muscle , sometimes is not expressed in fibroblasts

A mention to the "muscle phenotype" not found in fibroblasts has been added page 3.

Mention Acylcarnitines on blood as a diagnostic tool for FAOX

Acylcarnitines as diagnostic tool for FAO disorders is now mentioned page 2, 3rd paragraph.

2.1. Re  partial correction of  FAO flux with Bezafibrate (and other molecules);  Please relate-if corrected to 50% this would  actually be sufficient, since carriers have approximately ~ 50% activity and they are generally healthy

Concerning this point, we consider that healthy carriers with approximately 50% enzyme activity will present normal (100%) values of FAO flux. Therefore, conversely, an individual with 50% of normal FAO flux will be considered as deficient and most likely has a reduced enzyme activity i.e below 50%. This point has been addressed page 7.

2.2. Re patient trials; were the CPT2 VLCAD patient mutations known? Any correlation to fibroblasts results , carrying the same mutations?  

Concerning our clinical trial, we observed a positive response to bezafibrate in all the patients' myoblasts, whatever genotypes. For the second clinical trial (Orngreen et al), no patients' genotypes were reported which hampered any analysis. Finally, in the third trial, no FAO assessments were performed either ex vivo or in vivo. All these points are now mentioned page 14 last paragraph.

3. Consider mentioning the “French paradox”  

A mention to the "French paradox" has been added page 15.

6.2. Consider that complexes I and III are generally thought to be the major sites of  ROS production and its questionable if  defect in other complexes  would produce ROS      

This point has been addressed page 30.

Consider discussing data from PMID: 28093355

The article is now discussed page 31.

7. Mention that  PARP inhibitors are sometimes used in cancer treatment

The use of PARP inhibitors in cancer therapy has been addressed page 33-34.

Consider discussing data from PMID: 29886046

This reference is now discussed page 35.

It would be helpful to add another column- summarizing compounds and effects:  positive/negative/mixed (some parameters positive some negative) /not significant

Table 1 and 2: a new column has been added as requested by the reviewer.

All figures : There seem to be question marks   in the text boxes???.  Is this due to conversion to pdf??

We checked this point with the journal, and it appears that the original figures, sent and received, were fine, without any "?". They did not understand what the problem could be, and think that it might have something to do with the downloading process, which could have introduced these misprints. We hope that the problem will not happen again.

Round 2

Reviewer 2 Report

The authors revised the MS and responded to my comments. I have a minor comment: The study by Jang et al, 2017, PMID: 27604998) is now discussed in the revised MS (p.14), however, the reference is not shown in the References List. (shown as Juang 2017, see below)

p.14: "Taz mice were shown to exhibit progressive dilated cardiomyopathy with left ventricular dysfunction [134], and reduced exercise performance [133]. The knockdown of tafazzin induced inactivation and destabilization of RC supercomplexes (SC) and decreased gene expression of mitochondrial FAO enzymes and RC complexes in cardiac muscle [133]. More recently, Jang et al reported a more detailed characterization of heart dysfunction in Taz mice. In particular, the authors showed that in addition to the loss of SC integrity leading to decreased CI, CII and CIII activities, Taz heart showed high basal but low Ca++ -induced mitochondrial swelling and mitoROS, associated with high levels of cyclophilin D, a key regulator of mitochondrial permeability transition pore opening (juang 2017). After 4 months on a pelleted rodent chow containing 0.05% BZ, Taz mice exhibited improved systolic function, but intriguingly did not ameliorate their exercise capacity. Importantly, RNA-seq analysis revealed drug-induced up-regulation of several genes related to mitochondrial energy metabolism (FAO, RC, TCA) in the cardiac muscle [133]."